

# Modeling of Gas-Wall Partitioning of Organic Compounds Using a Quantitative Structure-Activity Relationship

Sanghee Han, Myoseon Jang, and Huanhuan Jiang

Department of Environmental Engineering Science, University of Florida, Gainesville, Florida, USA

*Correspondence to*: Myoseon Jang (mjang@ufl.edu)

**Abstract.** This study streamlines modeling of the gas–wall process (GWP) of semivolatile organic compounds (SVOC) by predicting gas–wall equilibrium partitioning constant ($K_{w,i}$) and accommodation coefficient ($\alpha_{w,i}$) of SVOC($i$) using a quantitative structure–activity relationship. PaDEL-Descriptor, software that calculates molecular descriptors, is employed to obtain physicochemical parameters (i.e., hydrogen bond acidity ($H_{d,i}$), hydrogen bond basicity ($H_{a,i}$), dipolarity/polarizability

($S_i$), and polarizability ($\alpha_i$)) of SVOC($i$). For the prediction of $K_{w,i}$, activity coefficients ($\gamma_{w,i}$) of SVOC($i$) to the chamber wall are semiempirically predicted using chamber data in the form of a polynomial equation coupled with the physicochemical parameters. $\gamma_{w,i}$ of various SVOCs differ in functionalities and molecular sizes ranging from $10^0$ to $10^4$. We conclude that the estimation of $\gamma_{w,i}$ is essential to improve the prediction of $K_{w,i}$. To predict the impact of relative humidity (RH) on GWP, each coefficient in the polynomial equation for $\ln(K_{w,i})$ was correlated to RH. Increasing RH enhanced GWP significantly

for all polar SVOCs. For example, the predicted $K_{w,i}$ of 1-heptanoic acid increased more than three times (from 0.58 to 1.96) by increasing RH from 0.4 to 0.75 due to the reduction in $\gamma_{w,i}$. The characteristic time for GWP are estimated using $K_{w,i}$ and $\alpha_{w,i}$ to evaluate the effect of GWP on secondary organic aerosol (SOA) mass. It might be significant in the absence of inorganic aerosol, but insignificant in the presence of electrolytic salts, where aqueous reactions dominate SOA growth.

## 1.  Introduction

Organic Aerosol (OA) can be produced via the atmospheric secondary process of reactive hydrocarbons which emitted from both vegetation and anthropogenic sources as well as the emission from primary sources such as fuel combustion, industries, and vehicles into the ambient air. Secondary organic aerosol (SOA), generated by atmospheric process of reactive hydrocarbons, constitutes a large proportion (up to 40%) of OA in the ambient air (Hallquist et al., 2009; Volkamer et al., 2006) and it can significantly impact on climate (Seinfeld and Pandis, 2016), visibility (Park et al., 2003), and human health (Cohen

et al., 2017). Thus, a large effort has given to the prediction of SOA budget in regional and global scales (Carlton et al., 2009; Volkamer et al., 2006; Barsanti et al., 2017). SOA models have been developed based on the mass balance of the hydrocarbon mass consumption during the atmospheric process. The prediction of SOA formation has been approached using a partitioning-based model with semiempirical parameters that are obtained from several semivolatile surrogate products (Odum et al., 1996; Donahue et al., 2006). Historically, the Community Multi-scale Air Quality Model (CMAQ) tends to predict SOA mass lower

than that observed because of missing precursors and reaction mechanisms (Barsanti et al., 2013), especially during the summer (Appel et al., 2017). Hence, significant efforts have been made to add in-particle chemistry into SOA models to allow the formation of nonvolatile oligomers in the aerosol phase (Jang et al., 2002; Cao and Jang, 2010; McNeill et al., 2012; Im et al., 2014; Beardsley and Jang, 2016; Zhou et al., 2019).

Recent studies have also reported that the underestimated SOA mass in the model prediction can be attributed to the chamber

because oxygenated products can partition on the reactor wall. Consequently, this negative bias in the model parameters can mislead modeling efforts to predict SOA formation at regional scales. McMurry and Grosjean (1985) was the first to address





the loss of gaseous compounds of very volatile organic species (i.e., *n*-butane, *n*-pentane, and toluene) and atmospherically important inorganic tracers (i.e., NO, NO₂, ozone, and SO₂) on the wall. The gas–wall process (GWP) of Semivolatile Organic Compounds (SVOCs) has been shown only recently to constitute a significant potential bias in atmospheric chamber studies

and has been noted as a source of substantial underestimation of SOA burdens. For example, Zhang et al. (2014) reported that the GWP can lead to a SOA yield lower by a factor of 1.1 to 4.2 and particularly, that of toluene was found from 2.1 to 4.2. La et al. (2016) indicated that the SOA yield inferred from a chamber study can be underrated by more than 50% in experiments and explicit model simulations for alkane and alkene series. To date, the influence of GWP on the predicted SOA yields remains controversial due to the limitations of the experimental database, which measures the vapor concentration. The accurate

measurement of vapor concentration in the initial point is burdensome, without the delay attributable to the time required to finish injecting organic species into a chamber.

The gas–wall partitioning coefficient ($K_{OM,i}$, m³ µg⁻¹) of SVOC($i$) to the absorbing organic matter ($OM_{wall}$) on the Teflon film and the mass accommodation coefficient ($\alpha_{w,i}$) are key parameters to predict GWP of various SVOCs. The unitless partitioning coefficient ($K_{w,i}$) is estimated by multiplying $K_{OM,i}$ by the concentration of $OM_{wall}$ (mg m⁻³). $K_{w,i}$ is influenced by

$OM_{wall}$, which can be altered by the chamber history (i.e., the fresh wall and the aged wall). The vapor pressure of SVOC($i$) ($p^{\circ}_{L,i}$) is frequently employed to predict multiphase partitioning of SVOCs. For example, a simple relationship between the gas-particle partitioning coefficient ($K_{p,i}$) of SVOCs and their saturated vapor pressures (Pankow, 1994; Im et al., 2014) was employed to predict $K_{p,i}$. $p^{\circ}_{L,i}$ is useful in predicting $K_{p,i}$ for a homologous series that is similar in activity coefficient ($\gamma_{p,i}$). However, the variation of $\gamma_{p,i}$ of various SVOCs in different functionalities can be large, ranging from $10^0$ to $10^3$ and it can

significantly influence the predictability of $K_{p,i}$ (Jang et al., 1997; Jang and Kamens, 1998; Beardsley and Jang, 2016). Similar to $K_{p,i}$, several studies have attempted to predict $K_{OM,i}$ using SVOC($i$)'s saturated concentration ($C_i^*$) calculated as:

$$C_i^* = \frac{MW_{OM}p^{\circ}_L\gamma_{w,i}}{RT} \tag{1}$$

where $MW_{OM}$ is the mean molecular weight of the organic matter on the Teflon film, $R$ is the gas constant (8.21×10⁻⁵ m³ atm

K⁻¹ mol⁻¹), and $T$ (K) is the temperature, and $p^{\circ}_{L,i}$ (atm) assuming that the activity coefficient ($\gamma_{w,i}$) of SVOCs to the $OM_{wall}$

is one (Krechmer et al., 2016). However, the $\gamma_{w,i}$ values of different SVOCs on the chamber wall can vary depending on SVOC's polarities and molecular size and influence of $K_{OM,i}$. SOA products originating from the oxidation of hydrocarbons are diverse in functionalities. For example, the oxygen to carbon (O:C) ratio of α-pinene products is 0.43 on average (Zhang et al., 2015a; Chen et al., 2011) but that of isoprene products is about 0.8 (Bertram et al., 2011; Chen et al., 2011; Kuwata et al., 2013). The variation of the O:C ratios of products is large even in an individual SOA. For example, the O:C ratios of 1,3,5-

trimethylbenzene SOA products range from 0.11 to 3.0 (Jenkin et al., 2012; Im et al., 2014; Zhou et al., 2019). The hygroscopicity of the organic-coated Teflon wall also can influence $\gamma_{w,i}$ (Krechmer et al., 2016; Huang et al., 2018). To improve the accuracy in the prediction of $K_{OM,i}$, the estimation of $\gamma_{w,i}$ is called for.

$\alpha_{w,i}$ is the fraction that reversible uptake of a gas-phase species will occur upon collision with the chamber wall. $\alpha_{w,i}$ can vary from 1 to below 10⁻⁷ depending on volatility of the species and compound of surface (Brune, 2019). The magnitude of

$\alpha_{w,i}$ is also disputably estimated semiempirically using chamber data as a range from $10^{-5}$ to $10^{-8}$ (Bian et al., 2015; Zhang et al., 2015b; Krechmer et al., 2016).

A quantitative structure–activity relationship (QSAR) approach was applied for the first time in this study to predict $K_{w,i}$ and $\alpha_{w,i}$. The QSAR theory was traditionally developed in the form of a linear solvation energy relationship (LSER) (Alan, 1992; Puzyn et al., 2010), by predicting the solvent effect on the rate or equilibrium constant of an analyte using the multi regression




equation associate with physicochemical parameters. The LSER-based prediction of a solvation property (SP) is typically expressed as:

$$\log SP = c + aH_{d,i} + bH_{a,i} + eE_i + sS_i + \cdots \tag{2}$$

where $H_{d,i}$, $H_{a,i}$, $E_i$, and $S_i$ indicate hydrogen bond acidity, hydrogen bond basicity, excess molar refraction, and dipolarity/polarizability, respectively (Abraham and McGowan, 1987; Abraham et al., 1991; Platts et al., 1999). The

physicochemical parameters are not limited to the terms in Eq. (2) and can be extended to additional parameters depending upon the properties of SP, series of solutes, and the medium. QSAR has also been applied widely in environmental studies, such as partitioning of an organic compound in airborne particles (Jang and Kamens, 1999; Arp and Goss, 2009; Endo and Goss, 2014).

In this study, the estimation of $\gamma_{w,i}$ was performed using the QSAR approach, which utilizes physicochemical parameters

such as $H_{d,i}$, $H_{a,i}$, $E_i$, $S_i$, and polarizability ($\alpha_i$) to improve the predictability of $K_{w,i}$. Each coefficient in the QSAR-based equation used to predict $K_{w,i}$ was correlated to relative humidity (RH) using chamber data under different RH levels. To assess GWP's significance, the characteristic time of GWP ($\tau_{GWP}$) calculated using $K_{w,i}$ and $\alpha_{w,i}$ was compared with the characteristic times of important atmospheric processes, such as gas-particles partitioning, gas-phase oxidation, and aerosol phase reactions.

## 2. Experiment

### 2.1. Chamber experiment

The University of Florida Atmospheric PHotochemical Outdoor Reactor (UF-APHOR) chamber is bisected into East and West chambers by an airtight door in the middle. Each chamber's volume is 52 m³ with an 86 m² surface area. Table 1 summarizes the experimental conditions of the UF-APHOR chamber data and the literature data (Matsunaga and Ziemann, 2010; Yeh and

Ziemann, 2015) that were used to develop the model to predict $K_{w,i}$ and $\alpha_{w,i}$ under different levels of RH. The experimental procedure is described in Sect. S1. To reduce the delay attributable to inject chemicals into the chamber, a cocktail of the organic mix (Table 2) was introduced to the chamber using the two chambers (Fig. S1). The East chamber was used to inject chemicals with a typical manifold heated under a clean air stream. Then, the organic vapor in the East chamber was transferred to the West chamber through the door between the two chambers for 10 minutes. The large size of UF-APHOR benefits the

increase in time to reach equilibrium. The impact of the surface area (A) to volume (V) ratio of the chamber on $K_{w,i}$ and GWP equilibrium time will be discussed in Sect. 4.5 and 4.6. The SVOC vapor was collected from the West chamber using a 40-cm 5-channel annular denuder immediately after closing the door between the two chambers (Sect. S1). The concentrations of SVOC vapor were corrected for the chamber dilution by monitoring the concentration of CCl₄ using gas chromatography with a flame ionization detector (GC-FID, Hewlett-Packard 5890, Palo Alto, CA, USA). To ensure particle formation, the particle

size distribution was monitored using a scanning mobility particle sizer (SMPS, TSI 3080, Shoreview, MN, USA) and a condensation particle counter (CPC, TSI 3022A, Shoreview, MN, USA).

### 2.2. Chemical analysis

The organic vapor was collected using a 40-cm 5-channel annular denuder coated with XAD-4 resin powder (Gundel et al., 1995; Leach et al., 1999). Details of the sampling and workup procedures, chemical purity, and analytical precision are reported

in Sect. S1 in SI. In brief, the denuder sample was extracted with dichloromethane. Extracted samples were concentrated with an evaporator (Heidolph Rotary Evaporator Laborota 4001, Schwabach, Germany) at 80°C. The SVOC's concentrations were





analyzed with a gas chromatography/mass spectrometer (GC/MS) (Varian 3800/2000 GC/MS, Palo Alto, CA, USA). Six deuterated polycyclic aromatic hydrocarbons (naphthalene-d$_8$, acenaphthylene-d$_{10}$, anthracene-d$_{10}$, fluoranthene-d$_{10}$, pyrene-d$_{10}$, and chrysene-d$_{12}$) were used as internal standards.

**2.3. Absorbing organic matter on the Teflon film and its hygroscopicity**

The functional-group's distribution of the nonvolatile organic matter adsorbed on the Teflon film ($M_{wall-OM}$) was analyzed using a Fourier Transform Infrared (FTIR) spectrometer (Nicolet Magma 560, Madison, WI, USA). $M_{wall-OM}$ was collected by wiping the Teflon film (surface area: 20 cm × 20 cm) with an acetone-drenched nylon filter. The solution extracted with 10 mL of acetone was concentrated by clean air and placed on a silicon FTIR disc (13 mm × 2 mm, Sigma-Aldrich, St. Louis,
MO, USA). The mass of collected $M_{wall-OM}$ was determined using the mass difference between the measure taken before and after placing $M_{wall-OM}$ on the FTIR disk by an analytical balance (Mettler Toledo MX5 microbalance, Columbus, OH, USA). The FTIR spectrum of $M_{wall-OM}$ (Fig. 1) is decoupled into the functional bends (i.e., C–H stretching, O–H stretching (water, alcohol, and carboxylic acid), C=O stretching (i.e., carboxylic acid and carbonyls), C=C stretching, H$_2$O bending, ester C–O) using a curve-fitting method assuming that each bend has a Gaussian shape (Li et al., 2016). The construction of the
functional composition was calculated using the peak area of each decoupled FTIR bend and the database of the relative intensity of various FTIR bends originating from various pure compounds.

**3.  Model Description**

The structure of the predictive model for $K_{w,i}$ and $\alpha_{w,i}$ is illustrated in Fig. 2. $K_{w,i}$ and $\alpha_{w,i}$ are predicted in the form of a polynomial equation coupled with QSAR descriptors using the experimental data that measured gas-phase concentrations of
SVOCs in the UF-APHOR chamber. The GWP is assumed as a partitioning of SVOCs to $OM_{wall}$, which consists of the organic composition of $OM_{wall}$ ($M_{wall-OM}$) and the water content ($M_{wall-water}$) associated with $M_{wall-OM}$. The permeability of SVOCs through the Teflon film is assumed to be negligible.

The Teflon film is highly crystalline with high density (2.15 kg m$^{-3}$) (Leivo et al., 2004). Thus, the Teflon film has low permeability to liquid, organic vapors, and moisture (Kaur et al., 2001). Teflon film is more hydrophobic than aliphatic
compounds. For example, the solubility parameter of Teflon film (FEP, Fluorinated Ethylene Propylene) based on Hansen's parameter is only 12.7, that of polyethylene film (or long-chain alkane) is 17.6, and that of ethylene glycol is 24.8 (Barton, 2017). Thus, wax-like long-chain alkanes presumably have a higher affinity to Teflon film than polar compounds. *n*-Alkanes are common species in both primary emissions (combustion of fossil fuels) and biogenic emissions (Simoneit, 1989; Bi et al., 2003; Lyu et al., 2019). These long-chain alkanes can enter the chamber during the ventilation process, and they can also
constantly intrude the chamber via chamber dilution. Based on FTIR spectrum (Fig. 1), $M_{wall-OM}$ is hydrophobic, showing a low O:C ratio. It seems that polar products that are created during chamber studies (i.e., SOA formation) and partition to the chamber wall degrade more quickly than wax-like alkanes under ambient sunlight. Oxygenated compounds such as aldehydes and conjugated alkenes are generally more labile for photolysis and atmospheric oxidants than long-chain alkanes. The chemical characteristics of $OM_{wall}$ will be discussed more in Sect. 4.1.

**3.1  Gas–wall process based on absorption–desorption kinetics**

SVOC(*i*)'s gas–wall partitioning is described by absorption–desorption kinetics and can be expressed as:

$$SVOC_{g,i} \underset{k_{off,i}}{\overset{k_{on,i}}{\rightleftarrows}} SVOC_{w,i} \qquad (3)$$





in which $k_{on,i}$ and $k_{off,i}$ are the SVOC's rate constants for deposition and evaporation (desorption), respectively. Based on a mass balance, SVOC's steady-state concentration in the gas phase ($C_{g,i}$, µg m$^{-3}$) is expressed as:

$$\frac{dC_{g,i}}{dt} = -k_{on,i}C_{g,i} + k_{off,i}C_{w,i} \qquad (4)$$

where $C_{w,i}$ is SVOC($i$)'s concentration (µg m$^{-3}$) on the Teflon wall. The sum of $C_{g,i}$ and $C_{w,i}$ is denoted as $C_{T,i}$. From the analytical solution of Eq. (4), $C_{g,i}$ can be expressed as follows:

$$C_{g,i} = \frac{k_{on,i}C_{T,i}}{k_{on,i}+k_{off,i}} e^{-(k_{on,i}+k_{off,i})t} + \frac{k_{off,i}C_{T,i}}{k_{on,i}+k_{off,i}} \qquad (5)$$

At equilibrium ($\frac{dC_{g,i}}{dt}=0$), SVOC($i$)'s $K_{OM,i}$ onto $OM_{wall}$ is described in the form of traditional gas-particle partitioning as

(Pankow, 1994):

$$K_{OM,i} = \frac{7.501RT}{10^{9}MW_{OM}\gamma_{w,i}p^{\circ}_{L,i}} = \frac{C_{w,i}}{C_{g,i}OM_{wall}} = \frac{k_{on,i}}{k_{off,i}OM_{wall}} \qquad (6)$$

$R$ is the ideal gas constant (8.314 J mol$^{-1}$ K$^{-1}$) and $T$ is the temperature (K). $MW_{OM}$ is $OM_{wall}$'s average molecular weight, and $p^{\circ}_{L,i}$ is liquid vapor pressure (mmHg) of SVOC($i$). $K_{w,i}$ is described as:

$$K_{w,i} = K_{OM,i}OM_{wall} = \frac{k_{on,i}}{k_{off,i}} \qquad (7)$$

By applying $K_{w,i}$ to Eq. (5), $C_{g,i}$ is rewritten as:

$$C_{g,i} = \frac{K_{w,i}C_{T,i}}{K_{w,i}+1} e^{-k_{on,i}\left(1+\frac{1}{K_{w,i}}\right)t} + \frac{C_{T,i}}{K_{w,i}+1} \qquad (8)$$

SVOC's diffusion from the gas phase to the Teflon wall was assumed to be the only significant mechanism of the vapor loss to the wall in a well-stirred chamber. $k_{on,i}$ is a fractional loss rate (McMurry and Grosjean, 1985) expressed as:

$$k_{on,i} = \left(\frac{A}{V}\right)\frac{\alpha_{w,i}\bar{v}_i/4}{1+\frac{\pi\alpha_{w,i}\bar{v}_i}{8(K_eD)^{1/2}}} \qquad (9)$$

$\bar{v}_i$ is the gas molecules' mean thermal speed. $D$ (1.0 × 10$^{-6}$ m$^2$ s$^{-1}$) and $K_e$ (0.12 s$^{-1}$) are referred to as the diffusion coefficient and coefficient of eddy diffusion applied as a fixed number, respectively. Increasing the A/V ratio can increase $k_{on,i}$ and thus $K_{w,i}$ suggesting that GWP is higher with a smaller chamber.

### 3.2 Prediction of $K_{w,i}$ and $\alpha_{w,i}$ using QSAR

By taking the logarithm, the theoretical $K_{w,i}$ is described as:

$$\ln(K_{w,i}) = -\ln(\gamma_{w,i}) - \ln(p^{\circ}_{L,i}) + \ln(\frac{7.501RTOM_{wall}}{10^{9}MW_{OM}}) \qquad (10)$$

In this study, the term $\ln(\gamma_{w,i})$ was predicted by the QSAR approach using the physicochemical descriptors $H_{d,i}$, $H_{a,i}$, $S_i$, $E_i$, and $\alpha_i$. Then, Eq. (10) can be rewritten as:

$$\ln(K_{w,i}) = -(a_pH_{d,i} + b_pH_{a,i} + s_pS_i + e_pE_i + r_p\alpha_i + c_p) - \ln(p^{\circ}_{L,i}) + \ln(\frac{7.501RTOM_{wall}}{10^{9}MW_{OM}}) \qquad (11)$$

For each compound, descriptors ($H_{d,i}$, $H_{a,i}$, $S_i$, $E_i$, and $\alpha_i$) were obtained using PaDEL-Descriptor, (Yap, 2011) and $p^{\circ}_{L,i}$ was





calculated with group contributions (Stein and Brown, 1994; Zhao et al., 1999).

The prediction of $\alpha_{w,i}$, which is dependent on SVOC's physicochemical properties and the chamber's characteristics (A/V), also can be approached by using QSAR as follows:

$$\ln(\alpha_{w,i}) = a_d H_{d,i} + b_d H_{a,i} + s_d S_i + e_d E_i + r_d \alpha_i + c_d \qquad (12)$$

$K_{w,i}$ and $\alpha_{w,i}$ were obtained in the form of a polynomial equation by fitting the calculated $C_{g,i}$ (Eq. (8)) to the observed

$C_{g,i}$. The significance of the coefficient associated with each QSAR parameter in Eq. (11) and Eq. (12) was determined based on the statistical values (Sect. S3). The estimation of the characteristic time ($\tau$) helps understand the role of $K_{w,i}$ and $\alpha_{w,i}$ in GWP for various SVOCs because $\tau_{GWP}$ indicates the way SVOC can reach equilibrium quickly. As shown in Eq. (5), $\tau_{GWP}$ can be calculated from $k_{on,i}$ and $k_{off,i}$ as follows:

$$\tau_{GWP} = \frac{1}{k_{on,i} + k_{off,i}} \qquad (13)$$

## 4.    Results and Discussion

### 4.1. Chemical characteristics of OM$_{wall}$

As seen in Fig. S2 in Sect. S2, the change in the functional-group distributions in $OM_{wall}$ is small year around. Therefore, the composition of $M_{wall-OM}$ is fixed as seen in Fig. 1. Table S1 shows the functional-group distribution of $M_{wall-OM}$. The O:C ratio of $M_{wall-OM}$ is 0.27 supporting that $M_{wall-OM}$ is hydrophobic. The amount of $OM_{wall}$ was set to 18.52 mg m$^{-3}$

by averaging its measurements. $M_{wall-water}$ was determined using FTIR spectra obtained by interfacing the FTIR spectrometer with a specially fabricated, RH-controlling flow tube (Fig. S3 in Sect. S2) (Jang et al., 2010; Beardsley et al., 2013; Zhong and Jang, 2014). To determine $MW_{OM}$, the number of functional groups within the constrained functional distribution is optimized with the UNIQUAC Functional-group Activity Coefficients (UNIFAC) model by fitting the activity coefficient of water in $OM_{wall}$ to RH. The elemental composition of $M_{wall-OM}$ is determined as $C_{15}O_4H_{24}$ using the

functional composition and $MW_{OM}$. A detailed description of the estimation of chemical characteristics of $M_{wall-OM}$ and $M_{wall-water}$ is found in Sect. S2.

### 4.2. Impact of humidity on GWP

The hygroscopicity of $OM_{wall}$ can impact on $K_{w,i}$ due to the changing $M_{wall-water}$ as a function of RH. To predict the influence of RH on $K_{w,i}$, each coefficient in the QSAR-based polynomial equation was correlated to RH using the chamber

data obtained under the three different RH levels. The two datasets obtained at RH = 0.40 (Oct 8, 2018) and RH = 0.75 (May 15, 2018) from the UF-APHOR chamber (this study) and the data under the dry condition (RH < 0.001) were obtained from the literature (Yeh and Ziemann, 2015; Matsunaga and Ziemann, 2010). The optimal coefficients of a multilinear regression model for $K_{w,i}$ and $\alpha_{w,i}$ were determined by the backward elimination procedure considering the adjusted $R^2$ and $p$-value. Table S2 summarizes the resulting linear regression coefficients for each experiment to predict $\ln(K_{w,i})$. Parameters $H_{d,i}$, $H_{a,i}$,

$S_i$, and $\alpha_i$ were chosen as independent variables for $K_{w,i}$ (Table S2). The $K_{p,i}$ was related exponentially to RH, as reported in previous studies (Jang and Kamens, 1998; Pankow et al., 1993). Fig. S4 illustrates the correlation between coefficient of each physicochemical parameter to predict $\ln(\gamma_{w,i})$ and RH. Thus, each coefficient in the polynomial equation for $\ln(K_{w,i})$ was correlated linearly to RH using the three sets of experimental data, as follows:

$$\ln(K_{w,i}) = (3.09RH - 5.17)H_{d,i} + (2.71RH - 2.80)H_{a,i} - (0.01RH + 0.19)\alpha_i + (1.73RH - 1.31)S_i + 9.00 - \ln(p^{\circ}_{L,i}) +$$

$$\ln\left(\frac{7.501RTOM_{wall}}{10^9 MW_{OM}}\right). \qquad (14)$$



The prediction is that $\ln(K_{w,i})$ was correlated positively with RH for QSAR descriptors $H_{a,i}$, $H_{d,i}$, and $S_i$ but not for $\alpha_i$. The $\ln(\frac{7.501RTOM_{wall}}{10^9 MW_{OM}})$ term slightly increased with increasing RH due to the variation in the $MW_{OM}$, calculated as a function of RH using hygroscopicity data (Fig. S3).

Fig. 3 illustrates the tendency of $C_{g,i}$ at the two different RH levels (0.4 and 0.75) over the course of the chamber experiment (May 15, 2018 and October 08, 2018). Fig. 4 shows $K_{w,i}$ prediction as a function of RH. The increasing RH enhanced $K_{w,i}$ significantly for all polar SVOCs. For example, $K_{w,i}$ of 1-heptanoic acid increased by a factor of three (from 0.58 to 1.96) when RH increased from 0.4 to 0.75. $K_{w,i}$ of the hydrophobic SVOC such as $n$-eicosane was, however insensitive to RH as reported by Huang et al. (2018).

Parameters $H_{d,i}$, $H_{a,i}$, $S_i$, and $\alpha_i$ were chosen to predict $\alpha_{w,i}$ based on the statistical analysis (Table S3) semiempirically using the three UF-APHOR data as:

$$\ln(\alpha_{w,i}) = -0.92H_{d,i} - 1.3H_{a,i} - 0.039\alpha_i - 0.98S_i - 10.69. \tag{15}$$

$\alpha_{w,i}$ decreased with increasing molecular size, a trend similar to that reported by Ye et al. (2016). In addition, $\alpha_{w,i}$ also decreased when $H_{d,i}$, $H_{a,i}$, and $S_i$ increased. The effect of RH on $\alpha_{w,i}$ was not derived in this study.

### 4.3. $K_{w,i}$, $\alpha_{w,i}$, and $\tau_{GWP}$ of various SVOCs: chamber data vs prediction

Values of SVOCs' $K_{w,i}$, $\alpha_{w,i}$, and $\tau_{GWP}$ were predicted and are summarized in Table 2 with the values of all the SVOCs' descriptors. $K_{w,i}$ was calculated using Eq. (14) at a given RH (0.75) and temperature (298 K) with 18.52 mg m$^{-3}$ of $M_{wall-OM}$ and 273.3 g mol$^{-1}$ of $MW_{OM}$ (Sect. 4.1 and S2). Overall, by reducing volatility, the SVOC has a greater $K_{w,i}$ with a large carbon number or strong hydrogen bonding. However, volatility alone cannot fully explain the tendency of $K_{w,i}$. For example, 1-decanoic acid's $K_{w,i}$ is nearly 7 times higher (12.80) than that of $n$-nonadecane (1.93), although $n$-nonadecane's estimated $p_{L,i}^\circ$ (1.09×10$^{-3}$ mmHg) is close to that of 1-decanoic acid (1.18×10$^{-3}$ mmHg). In general, a longer $\tau_{GWP}$ was found for SVOCs with high $K_{w,i}$ and low $\alpha_{w,i}$. For example, $n$-eicosane's $\tau_{GWP}$ (95 min) was significantly higher than that of 2-heptanol (21 min), while $n$-eicosane and 2-heptanol had a similar $\alpha_{w,i}$ (2.9 × 10$^{-6}$ and 3.3 × 10$^{-6}$, respectively). The dissimilar $K_{w,i}$ values of $n$-eicosane and 2-heptanol (1.93 and 0.28, respectively) occurred with large differences in $\tau_{GWP}$. The volatile (small $K_{w,i}$) and hydrophobic SVOCs can reach equilibrium quickly (short $\tau_{GWP}$). However, the predicted $C_{g,i}$ of citral deviated significantly from our observations. Aldehydes generally are more reactive than ketones because aldehydes can be oligomerized in the aerosol's aqueous layer or the wall (Jang and Kamens, 2001; Jang et al., 2002; Tong et al., 2006; Liggio and Li, 2006). Citral, a conjugated aldehyde, was reactive on the wall, and thus was excluded from the derivation of the QSAR model used to predict $K_{w,i}$ and $\alpha_{w,i}$.

The $\tau_{GWP}$ values obtained from this study ranged from 21 to 144 minutes (Table 2). Overall, the $\tau_{GWP}$ values obtained in this study are greater than those reported previously (Zhang et al., 2015b; Krechmer et al., 2016). For example, the $\tau_{GWP}$ of 2-tridecanone of this study was 71 minutes, while that in Yeh and Ziemann (2015) reported as less than 35 minutes. The difference in $\tau_{GWP}$ between laboratory studies can be explained with the difference in RH (0.75 at UF-APHOR vs 0.001 at Ziemann's chamber (Yeh and Ziemann, 2015) and the A/V ratio (i.e., 1.65 for the UF-APHOR chamber and 3.0 at Ziemann's chamber). The A/V ratio originating from the chamber dimension affects the calculation of $k_{on,i}$ as seen in Eq. (9) as well as the estimation of $OM_{wall}$. In addition, increasing RH can increase $\tau_{GWP}$ for polar compounds because the lower $\gamma_{w,i}$ (or elevated $K_{w,i}$) at the higher RH increases the time required to reach equilibrium. $k_{on,i}$ associated with SVOC flux to the chamber wall is not (or little) affected by RH.



### 4.4. Impact of $\gamma_{w,i}$ on the prediction of $K_{w,i}$

The observed $K_{w,i}$ are plotted against predicted $1/C_i^*$ (estimated from $RT/MW_{OM}\gamma_{w,i}p_L^\circ$ with unity $\gamma_{w,i}$) and $K_{w,i}$ in Fig. 5(a) and Fig. 5(b), respectively. Fig. 5 was constructed using the three datasets in this study and the data reported in the literature (Matsunaga and Ziemann, 2010; Yeh and Ziemann, 2015). When observed $K_{w,i}$ is plotted to $1/C_i^*$ (Fig 5(a)), a regression line with some scatter was obtained ($R^2 = 0.12$, except aldehydes). However, the inclusion of $\gamma_{w,i}$ (Fig 5(b)), as calculated from the QSAR method, dramatically improves the predictability of $K_{w,i}$ ($R^2 = 0.64$, except aldehydes). To demonstrate the potential impact of $\gamma_{w,i}$ on $K_{w,i}$, $1/C_i^*$ was plotted vs the predicted $K_{w,i}$ in Fig. 5(c). For various organic compounds that differ in functionalities and molecular sizes, $\gamma_{w,i}$ ranges $10^0$–$10^4$. Overall, the $K_{w,i}$ values of monofunctional oxygenated compounds, such as $n$-alcohols, 1-carboxylic acids, ketones, and $n$-alkanes reasonably accord with the predicted $K_{w,i}$ using Eq. (14). However, the prediction of diols and aldehydes deviated from observation by one order of magnitude (Fig. 5(b)).

The study of multifunctional SVOCs' (e.g., diols) GWP generally would be more difficult than that of monofunctional compounds because of the experimental difficulty of injecting multifunctional SVOCs into the chamber and the detection techniques. In addition, estimating multifunctional alcohols' $p_{L,i}^\circ$ could be problematic because of intramolecular hydrogen bonding's contribution when estimating their boiling point. Estimating multifunctional alcohols' descriptors may also be uncertain because of the lack of a database (Miller, 1990; Yap, 2011).

The vapor fraction ($F_{g,i} = C_{g,i}/C_{T,i}$) of the total concentration of SVOC($i$) at equilibrium can be calculated from $K_{w,i}$ with the following equation:

$$F_{g,i} = \frac{1}{1+K_{w,i}} \tag{16}$$

Fig. S5 shows the model predicted $F_{g,i}$ and experimentally measured $F_{g,i}$ using the same dataset applied to $K_{w,i}$. The smaller $F_{g,i}$ indicates the significance of GWP on the chemical loss to the chamber wall.

### 4.5. Characteristic time analyses of major atmospheric processes

To assess the importance of GWP on SOA formation, the $\tau_{GWP}$ predicted using $K_{w,i}$ and $\alpha_{w,i}$ was compared with the characteristic time ($\tau$) associated with other important atmospheric processes. Table S4 summarizes equations to estimate $\tau_{GWP}$, $\tau_{OH}$ (reaction with an OH radical at $[OH] = 1.33\times 10^7$ molecules cm$^{-3}$, at noon during a typical chamber experiment), $\tau_{or}$ (reaction in the organic aerosol phase), $\tau_{in}$ (reaction in the inorganic aqueous phase), and $\tau_{GP}$ (SVOC's gas-particle partitioning). The characteristic times ($\tau_{GWP}$, $\tau_{OH}$, $\tau_{or}$, $\tau_{in}$, and $\tau_{GP}$) are estimated in Fig. 6 for the eight surrogate SVOCs differentiated by two different levels of $K_{w,i}$ ($10^{-1}$ and $10^1$), $\alpha_{w,i}$ (($1 \times 10^{-6}$ and $4 \times 10^{-6}$), and the reactivity scale (fast and no reaction) in the aerosol phase (Im et al., 2014; Beardsley and Jang, 2016; Zhou et al., 2019).

To evaluate the significance of GWP on chamber-generated SOA mass, Figure 6 also provides $F_{g,i}$ along with the characteristic times of various processes of various SVOCs. Overall, GWP is significant with inert oxygenated products with large $K_{w,i}$ and large $\alpha_{w,i}$. $\tau_{GP}$ is as short as ~$1\times 10^1$ s and it is always shorter than $\tau_{GWP}$. $\tau_{GWP}$ increases by an order of magnitude by increasing $K_{w,i}$ from $10^{-1}$ to $10^1$. The large $\alpha_{w,i}$ ($4 \times 10^{-6}$) shortens $\tau_{GWP}$ compared with the small $\alpha_{w,i}$ ($1 \times 10^{-6}$) at given conditions. $\tau_{OH}$ is ~ $4\times 10^3$ s under the given conditions. $\tau_{GWP}$ is shorter than $\tau_{OH}$ when $K_{w,i}$ is small ($10^{-1}$). Both $\tau_{or}$ and $\tau_{in}$ change with the reactivity scale in the aerosol phase by six orders of magnitude (fast (A–D) vs. no reactivity (E–H) in Fig. 6). When the reactivity scale is fast, $\tau_{GWP}$ is significantly larger than both $\tau_{or}$ and $\tau_{in}$. When



oxygenated products in the phase are dominated by heterogeneously reactive organic species or hygroscopic inorganic electrolytes are present in the aerosol, GWP may be less influential on SOA formation.

In addition, $\tau_{GP}$, $\tau_{OH}$, $\tau_{GWP}$, $\tau_{or}$, and $\tau_{in}$ of oxygenated products that are found in SOA (toluene, α-pinene, or isoprene) are also estimated in Table S5. The products in Table S5 are chosen because of their high concentrations obtained in simulations using the explicit gas mechanisms (i.e., Master Chemical Mechanism (MCM) version 3.3.1 (Jenkin et al., 2012)) under the low NOx condition (HC ppbC/NOx ppb = 6.5 (toluene), 20 (isoprene), and 6.9 (α-pinene). $\tau_{or}$ and $\tau_{in}$ of the products in Table S5 were estimated using the rate constants obtained from the UNIPAR model (Im et al., 2014; Beardsley and Jang, 2016; Zhou et al., 2019). SVOCs are classified by reactivity in the aerosol phase in UNIPAR: very fast (VF, more than two aldehydes or epoxides), fast (F, at least two aldehydes or epoxides), medium (M, one aldehyde or epoxide), slow (S, ketones), and no reaction (P). For most toluene products, $\tau_{in}$ is much shorter than those of $\tau_{GWP}$ suggesting that GWP's influence on the formation of toluene SOA might be trivial in the presence of the electrolytic aqueous phase. However, its effect on SOA formation becomes significant in the absence of the inorganic aerosol based on $\tau_{GWP}$ and $\tau_{or}$ (Table S5).

Glyoxal, a major product from the photooxidation of toluene, has a high $p_{L,i}^{\circ}$ but can contribute to SOA formation significantly because of its high reactivity (Fu et al., 2008). As shown in Table S5, both glyoxal's $\tau_{or}$ and $\tau_{in}$ are much shorter than $\tau_{GWP}$ and thus, GWP has little effect. Of equal importance is the impact of GWP on toluene SOA, which would be less significant than the prediction of previous studies (Zhang et al., 2014; La et al., 2016) because of the contribution of reactive SVOCs' heterogeneous reactions in the aerosol phase, particularly in the electrolytic aqueous phase. The importance of the reaction of an SVOC with an OH radical is comparable to that of GWP. IEPOXB, an isoprene oxidation product, is known to contribute significantly to isoprene SOA (Hu et al., 2015). Similar to toluene SOA products, the GWP of IEPOXB's influence on isoprene SOA may not be important in the presence of an aqueous inorganic seed ($\tau_{GWP}$ ~3.3 h; $\tau_{in}$ ~0.1 h; $\tau_{or}$ ~5.8 h). The volatility of such α-pinene oxidation products as pinic acid, pinonic acid, and pinonaldehyde is relatively low compared with that of toluene SVOCs. Pinic acid, which is not reactive in the aerosol phase, can contribute quickly to SOA growth, but GWP will influence it in the long term. Pinonic acid's reactivity (slow) and pinonaldehyde's (medium) are comparably weak, but their low volatility and short $\tau_{in}$ ($\tau_{or}$) can lead SOA to form quickly without GWP's significant impact.

### 4.6. Model sensitivity and uncertainties

To determine the model sensitivity (%), the $F_{g,i}$ prediction for the major variables was performed by changing the values of RH (0.001 < RH < 0.8), temperature (288–308 K), OM$_{wall}$ (6.9 to 37.5 mg m$^{-3}$), and the A/V ratio (from 1 to 3) selectively. In Fig. 7(a) and (b), the modulation of $F_{g,i}$ for the two SVOCs (1-decanoic acid and $n$-nonadecane) was scaled to the standard condition (RH = 0.50, temperature = 298 K, $OM_{wall}$ = 18.52 mg m$^{-3}$, and A/V ratio =1.65 based on the possible range of the conditions). At the standard condition, the predicted $F_{g,i}$ of $n$-nonadecane and 1-decanoic acid are 0.23 and 0.17, respectively. Among the four variables, RH had the greatest influence on $F_{g,i}$, particularly for the polar compound (1-decanoic acid). Both 1-decanoic acid and $n$-nonadecane have similar $p_{L,i}^{\circ}$ (Table 2), but their sensitivity to RH differs. RH's effect on 1-decanoic acid is related to $K_{w,i}$ exponentially, as seen in Eq. (14). As shown in Fig. 7, the chamber dimension (the A/V ratio) and OM$_{wall}$ also influenced $F_{g,i}$ significantly. The A/V ratio increases $K_{w,i}$ by increasing $k_{on,i}$ (Eq. (9)) and impacts on the estimation of OM$_{wall}$. Thus, the A/V ratio can affect $F_{g,i}$ more than OM$_{wall}$.

Fig. 7(c) and (d) illustrates the uncertainty in $F_{g,i}$ attributable to the uncertainties of the major model parameters used to predict $K_{w,i}$ and $\alpha_{w,i}$: four descriptors ($S_i$, $\alpha_i$, $H_{d,i}$, and $H_{a,i}$) and $p_{L,i}^{\circ}$ in Eq. (14) and Eq. (15). Jover et al. (2004) reported the uncertainty associated with descriptors $H_{d,i}$, $H_{a,i}$, and $S_i$ as 8.4%, 3.6%, and 3.1%, respectively. The error associated with



$\alpha_i$ was estimated to be 5% based on other descriptors' errors and the database used for the additive method (Miller, 1990). Stein and Brown (1994) and Zhao et al. (1999) reported that the uncertainty associated with the group contribution method used for $p_{L,i}^{\circ}$ estimation is a factor of 1.45. Among the five model parameters in Fig. 7(c) and (d), $p_{L,i}^{\circ}$ was most influential on uncertainties of $F_{g,i}$.

The uncertainties of the QSAR model equations to predict $K_{w,i}$ and $\alpha_{w,i}$ can be determined from the experimental data. The time consumed for transferring organic vapors from the East chamber to the West chamber can impact the uncertainty in measurements of $C_{g,i}$. For most of the organic vapors, $\tau_{GWP}$ is significantly longer than 10 minutes (600 s). Therefore, the impact of the uncertainties of $C_{g,i}$ on the QSAR equations, which are attributed to various SVOCs, could be small.

### 4.7. Atmospheric implications

The QSAR-based model platform in this study allows the GWP for a variety of SVOCs' formation during the atmospheric processes of hydrocarbons to be predicted. We concluded that the $\gamma_{w,i}$ expressed with QSAR descriptors is essential to predict $K_{w,i}$, although $p_{L,i}^{\circ}$ is the most critical. Overall, GWP appears to be more significant with the more polar and larger SVOCs (section 4.3). Among the variables (temperature, RH, A/V ratio, and $OM_{wall}$), RH was the most influential in predicting $K_{w,i}$ (or $F_{g,i}$) for polar compound (in Sect. 4.6) and the A/V ratio also influenced $F_{g,i}$ significantly due to the impact of the A/V ratio on $OM_{wall}$. The accuracy of the data and the collected chemical species influence the coefficients' suitability in the QSAR-based model. The suitability of the QSAR-based model to predict simple monofunctional organic species' $K_{w,i}$ and $\alpha_{w,i}$ was examined, but it needs to be extended to accommodate various multifunctional organics.

The relative importance of GWP to form SOA was assessed by comparison of $\tau_{GWP}$ with $\tau_{GP}$, $\tau_{OH}$, $\tau_{or}$, and $\tau_{in}$ (in Sect. 4.5). $\tau_{GWP}$ is comparable to $\tau_{OH}$, although their values depend on SVOC's molecular structure as well as its reactivity toward an OH radical. In the presence of an aqueous inorganic seed (10 μg m$^{-3}$ of effloresced inorganic seed), reactive SVOCs' $\tau_{in}$ (Table S5) is significantly shorter than that of $\tau_{GWP}$ by six orders of magnitude, suggesting that GWP's role in SOA growth may not be significant. In the absence of an inorganic seed (or in the presence of a dry seed), $\tau_{or}$ and $\tau_{in}$ are comparable to $\tau_{GWP}$ for many SVOCs. However, $\tau_{or}$ decreases for highly reactive organics (i.e., glyoxal) in aerosol or volatile SVOCs (i.e., pinonic acid and pinonaldehyde, $p_{L}^{\circ} < 10^{-3}$ mmHg). Many studies have found a high fraction of oligomeric matter in SOA mass, (Jang et al., 2002; Tolocka et al., 2004; Gao et al., 2004; Beardsley and Jang, 2016; Lemaire et al., 2016), evidence that heterogeneous chemistry of organic species is an important mechanism by which SOA forms. For example, Lemaire et al. (2016) reported 50% of the oligomeric mass of isoprene SOA. If the inorganic aerosol is acidic, oligomerization is accelerated and $\tau_{in}$ becomes even shorter than the value shown in Table S5. For example, Beardsley and Jang (2016) reported that at least 65% of total SOA is oligomer, and an oligomer can constitute 85% of SOA in the presence of an acidic seed. Because of the attribution of aqueous chemistry in aerosol, the bias in SOA mass by GWP could be varied. Inert organic products' wall deposition also can influence SOA mass by modulating the formation of reactive SVOCs and the quantity of oxidants (i.e., OH radicals and ozone), and consequently affect the SOA mass.

As shown in Fig. 3(f), the product with an aldehyde group might be affected by heterogeneous reactions on the chamber wall and cause a bias in the estimation of SOA mass. The heterogeneous reactions of reactive aldehydes can be significant when electrolytic inorganic species build up on the Teflon wall. However, noticing that $\tau_{GP}$ is typically much shorter than $\tau_{GWP}$, the heterogeneous reaction of the reactive organic species in the aerosol phase should be much more significant than that on the wall surfaces when the reaction is irreversible.

To date, GWP has been studied by measuring individual SVOCs. However, there is a gap between GWP studies and their



application to conventional SOA models that are approached by an apparent gas-particle partitioning with the semiempirical parameters of only a small number of semivolatile surrogate products (Odum et al., 1996; Donahue et al., 2006). The model equations used to predict the GWP parameters ($K_{w,i}$, $k_{on,i}$, and $k_{off,i}$) in this study can be coupled with explicit SOA models. For example, the UNIPAR model predicts SOA mass by counting the volatility and aerosol phase reactions of the lumping species sourced from the SVOCs that are produced explicitly during hydrocarbons' photooxidation (Im et al., 2014; Beardsley

and Jang, 2016; Zhou et al., 2019). The integration of GWP parameters with the explicit SOA model improves the quantification of the bias in SOA mass attributable to GWP. GWP's predictability also improves the study of the gaseous SVOCs' degradation because numerous chemical species' degradation kinetics rely on chamber data. For example, Bertrand et al. (2018) reported that biomass burning markers' decay in particulate matter was not attributable to reactions with OH radicals, but gas-particle partitioning and GWP dominated it instead.

*Author contribution.* MJ designed the experiments and MJ, SH, and HJ carried them out. SH prepared the manuscript with contributions from all co-authors.

*Acknowledgments.* This research was supported by the awards from the NSF (AGS 1923651) and the National Strategic

Project-Fine particle of the National Research Foundation of Korea (NRF) funded by the Ministry of Science and ICT (MSIT), the Ministry of Environment (ME), and the Ministry of Health and Welfare (MOHW) (2017M3D8A1090654).



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





**Table 1. Chamber experiments used to model the gas-wall process of SVOCs.**

| Index (Fig. 5) | Date | RH | Temp (°C) | Dilution factor [a] |
|---|---|---|---|---|
| a | October 8, 2018 | 0.40 | 297~301 | $y = 0.38e^{0.049t}$ |
| b | May 15, 2018 | 0.75 | 294~296 | $y = 0.60e^{0.076t}$ |
| c | August 6, 2018 | 0.53 | 297~301 | $y = 0.40e^{0.040t}$ |
| d | (Yeh and Ziemann, 2015) | <0.001 | 297~299 | |
| e | (Matsunaga and Ziemann, 2010) | | | |

[a]      The factor for chamber dilution is calculated by measuring $CCl_4$ with GC-FID





**Table 2. Chemicals injected into the chamber and their physicochemical parameters (at 298 K) predicted using PaDel-Descriptor and estimated $K_{w,i}$, $\alpha_{w,i}$, and $\tau_{GWP}$ under a given condition (298K, RH=0.75).**

| No. | Chemical | MW | Molecular descriptors [a] | | | | | $p_L^\circ$ [b] | $K_{w,i}$ | $\alpha_{w,i}$ ($\times 10^{-6}$) | $\tau_{GWP}$ (s) |
| | | | $H_{d,i}$ | $H_{a,i}$ | $E_i$ | $S_i$ | $\alpha_i$ | | | | |
|---|---|---|---|---|---|---|---|---|---|---|---|
| 1 | 1-hexanoic acid | 116 | 0.59 | 0.41 | 0.15 | 0.51 | 14.08 | 0.078 | 1.12 | 2.7 | 3462 |
| 2 | 1-heptanoic acid | 130 | 0.59 | 0.41 | 0.15 | 0.51 | 16.26 | 0.026 | 2.18 | 2.5 | 4900 |
| 3 | 1-octanoic acid | 144 | 0.59 | 0.41 | 0.15 | 0.51 | 18.45 | 0.009 | 4.13 | 2.3 | 6276 |
| 4 | 1-decanoic acid | 172 | 0.59 | 0.41 | 0.15 | 0.51 | 22.82 | 0.001 | 12.80 | 1.9 | 8630 |
| 5 | Benzoic acid | 122 | 0.59 | 0.46 | 0.78 | 0.93 | 7.52 | 0.020 | 15.73 | 2.2 | 7171 |
| 6 | Pyruvic acid | 88 | 0.43 | 0.68 | 0.29 | 0.88 | 8.48 | 0.256 | 1.33 | 1.9 | 4244 |
| 7 | 2-heptanol | 116 | 0.35 | 0.40 | 0.19 | 0.41 | 17.49 | 0.317 | 0.28 | 3.3 | 1281 |
| 8 | 1-octanol | 130 | 0.35 | 0.39 | 0.21 | 0.45 | 19.68 | 0.109 | 0.53 | 3.0 | 2220 |
| 9 | 1-nonanol | 144 | 0.35 | 0.39 | 0.21 | 0.45 | 21.86 | 0.036 | 1.05 | 2.7 | 3566 |
| 10 | Benzyl alcohol | 108 | 0.35 | 0.56 | 0.83 | 0.88 | 8.75 | 0.093 | 4.77 | 2.4 | 5716 |
| 11 | Phenol | 94 | 0.55 | 0.43 | 0.83 | 0.88 | 6.56 | 0.988 | 0.44 | 2.6 | 1938 |
| 12 | 2,5-dimenthylphenol | 122 | 0.55 | 0.43 | 0.85 | 0.83 | 10.93 | 0.123 | 1.48 | 2.3 | 4390 |
| 13 | 2,6-dimethoxyphenol | 154 | 0.04 | 0.284 | 0.74 | 1.36 | 14.76 | 0.041 | 9.47 | 2.2 | 7335 |
| 14 | n-heptadecane | 240 | 0.00 | 0.08 | 0.04 | 0.13 | 39.36 | 0.005 | 0.76 | 3.8 | 2875 |
| 15 | n-nonadecane | 308 | 0.00 | 0.08 | 0.04 | 0.13 | 43.73 | 0.001 | 1.47 | 3.2 | 4771 |
| 16 | n-eicosane | 324 | 0.00 | 0.08 | 0.04 | 0.13 | 45.91 | 0.001 | 1.93 | 2.9 | 5700 |
| 17 | 2-Dodecanone | 184 | 0.00 | 0.42 | 0.18 | 0.60 | 27.19 | 0.053 | 0.63 | 2.5 | 3076 |
| 18 | 2-Tridecanone | 198 | 0.00 | 0.42 | 0.18 | 0.60 | 29.38 | 0.022 | 0.97 | 2.3 | 4235 |
| 19 | Decanal | 156 | 0.00 | 0.42 | 0.15 | 0.51 | 22.82 | 0.193 | 0.41 | 2.9 | 2018 |
| 20 | Citral | 152 | 0.00 | 0.46 | 0.15 | 0.51 | 18.45 | 0.142 | 1.31 | 2.8 | 3928 |
| 21 | Benzaldehyde | 106 | 0.00 | 0.47 | 0.15 | 0.51 | 7.52 | 0.982 | 1.66 | 3.3 | 3596 |

[a] Based on the QSAR approach with PaDEL-Descriptor (Yap, 2011).
[b] Calculated through group contribution (Zhao et al., 1999; Stein and Brown, 1994).


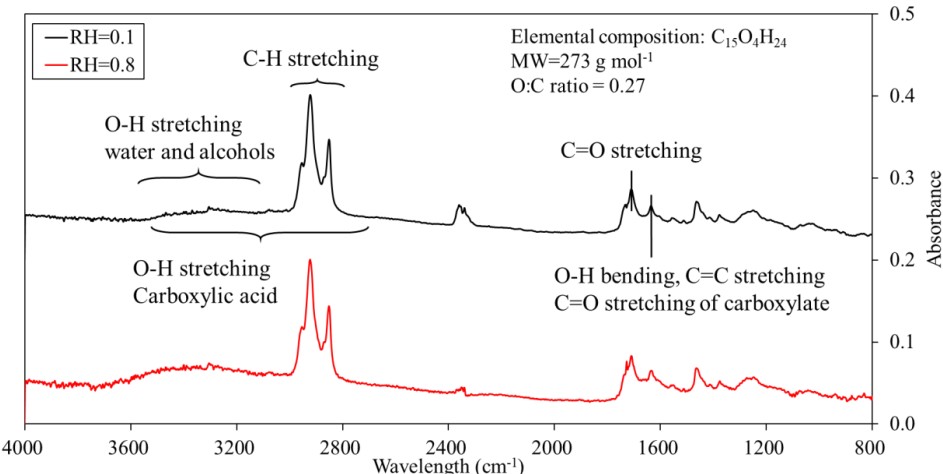

**Figure 1. FTIR spectra of the absorbing matter on the Teflon film. The absorbing matter was extracted with acetone and concentrated on a silicon disc. The FTIR spectra were obtained at the two different humidity (RH=0.1 and 0.8).**





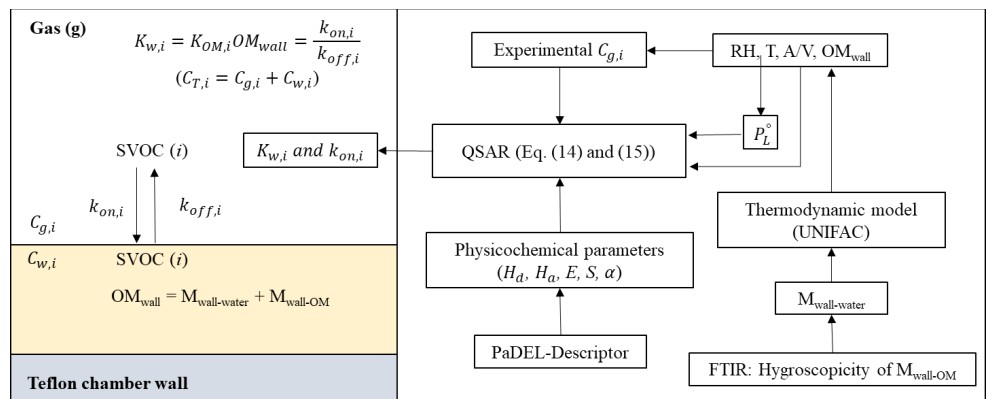

**Figure 2. Structure of the gas-wall partitioning model for various SVOCs.**





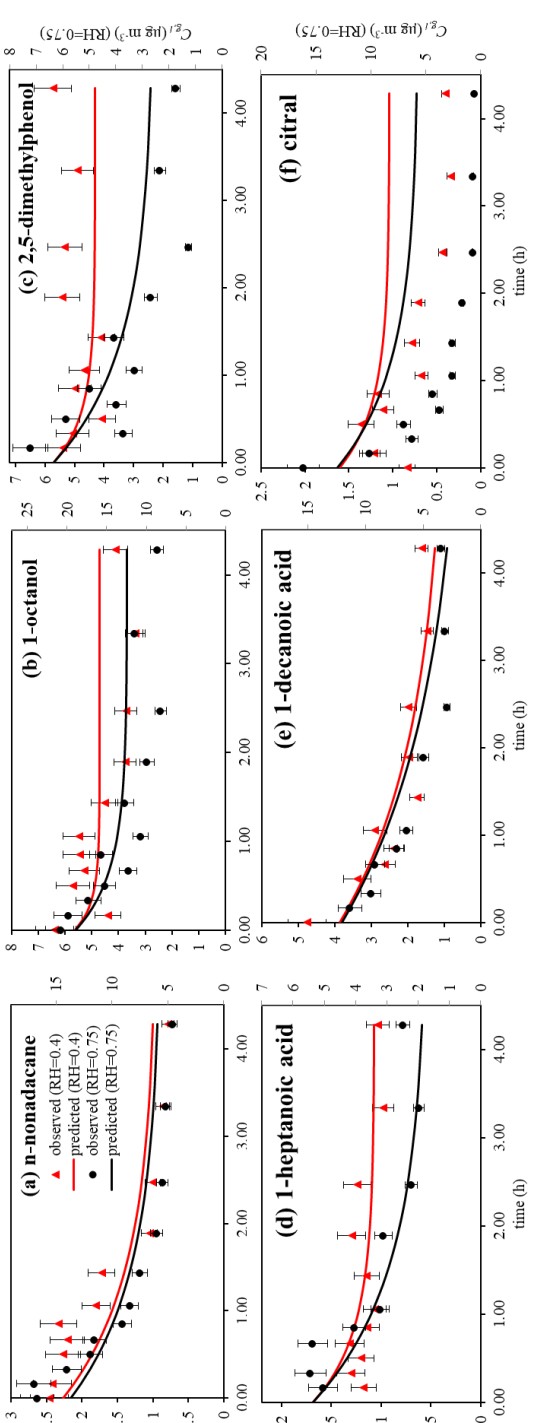

**Figure 3.** The observed concentrations of gaseous SVOCs ($C_{g,i}$ plot) in the chamber air for the two experiments at two different humidity (October 8, 2018 at RH=0.40 and May 15, 2018 at RH= 0.75) and the predicted $C_{g,i}$ (line) as a time series. The error bar associate with $C_{g,i}$ is estimated using standard deviation of the measured concentrations of internal standard (7–11 %).





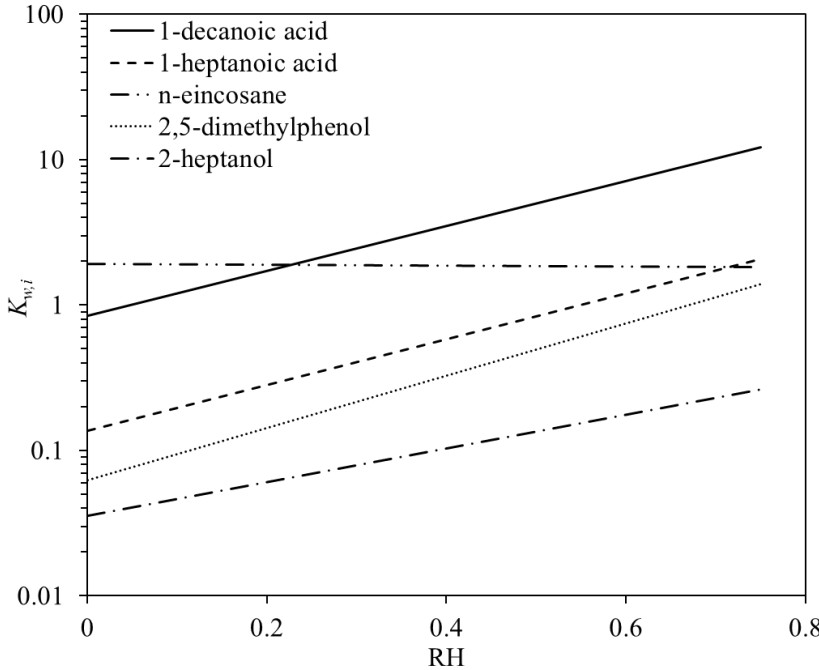

**Figure 4. The predicted $K_{w,i}$ with the QSAR-based GWP model as a function of RH.**



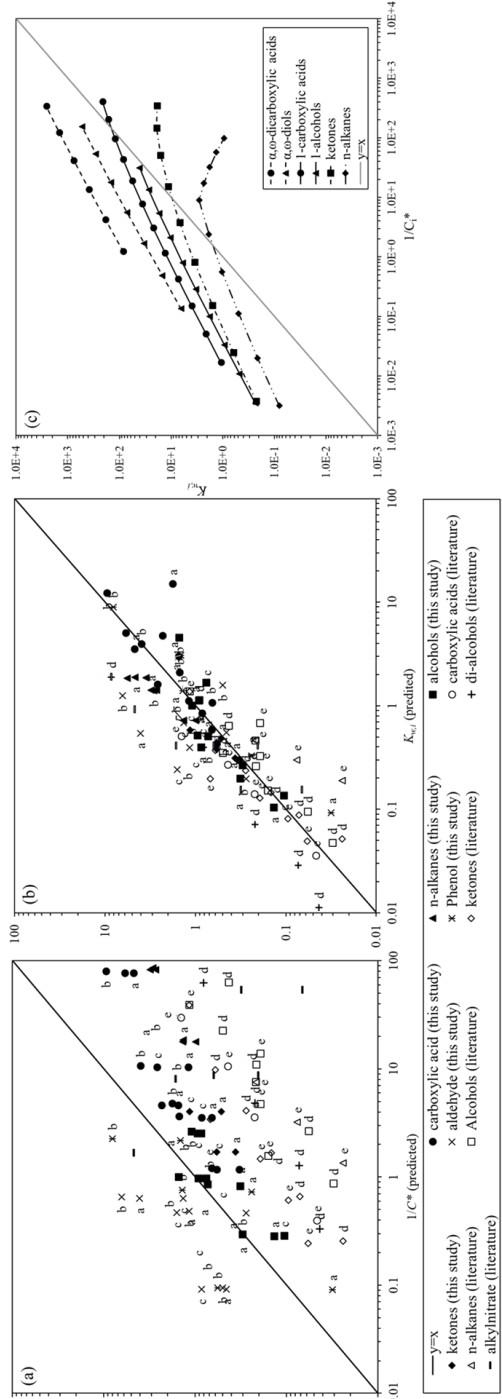

**Figure 5.** The observed $K_{w,i}$ is plotted vs the estimated $1/C_i^*$ (a), activity coefficient = 1) and the predicted $K_{w,i}$ (b) using activity coefficients estimated with the QSAR model (Eq. (14)) for various organic compound data sets from our study and literatures. The labels in (a) and (b) indicate the experiment summarized in Table 1 under different humidity conditions ($RH_a$=0.4, $RH_b$= 0.53, $RH_c$= 0.75, $RH_d$=$RH_e$=0.001). The experimental data set of alkylnitrate compounds is additionally obtained from Yeh and Ziemann (2014). The predicted $K_{w,i}$ is also plotted to $1/C_i^*$ for various SVOCs in different functionalities ((c), $n$-alkanes ($C_{2n}$, $n$=6–15), ketones ($C_{2n}$, $n$=4–13), 1-alcohols ($C_{12}$-$C_{16}$), 1-carboxylic acids ($C_5$-$C_{11}$), $\alpha$,$\omega$-diols ($C_6$-$C_{17}$), and $\alpha$,$\omega$-dicarboxylic acids ($C_2$-$C_7$)). This Figure shows that the $\gamma_{w,i}$ of SVOCs can influence the prediction of gas-wall partitioning in the order of the 4 (up to $10^4$).





| SVOC | A | B | C | D | E | F | G | H |
|---|---|---|---|---|---|---|---|---|
| $K_{w,i}$ | $10^{-1}$ | $10^{-1}$ | $10^{1}$ | $10^{1}$ | $10^{-1}$ | $10^{-1}$ | $10^{1}$ | $10^{1}$ |
| $\alpha_{w,i}$ | $1 \times 10^{-6}$ | $4 \times 10^{-6}$ | $1 \times 10^{-6}$ | $4 \times 10^{-6}$ | $1 \times 10^{-6}$ | $4 \times 10^{-6}$ | $1 \times 10^{-6}$ | $4 \times 10^{-6}$ |
| Reactivity scale | Fast | Fast | Fast | Fast | No reactivity | No reactivity | No reactivity | No reactivity |
| $F_{g,i}$ | 0.91 | 0.91 | 0.09 | 0.09 | 0.91 | 0.91 | 0.09 | 0.09 |

**Figure 6.** The characteristic time of the gas phase reaction of SVOC with an OH radical ($\tau_{OH}$), gas-particle partitioning ($\tau_{GP}$), gas-wall partitioning ($\tau_{GWP}$), the reaction of SVOC in the organic phase ($\tau_{or}$), and the reaction of SVOC in the inorganic phase ($\tau_{in}$) are illustrated at the bottom (see Table S4). $K_{w,i}$ and $\alpha_{w,i}$ influence $\tau_{GWP}$ and reactivity scale influences $\tau_{or}$ and $\tau_{in}$. The reactivity follows the class of the lumping in UNIPAR model (Fast and No reactivity). (Im et al., 2014; Beardsley and Jang, 2016; Zhou et al., 2019). The value on the bar chart indicates $\tau_{GWP}$. The propagation error is calculated based on the uncertainties associated with parameters (Table S4).



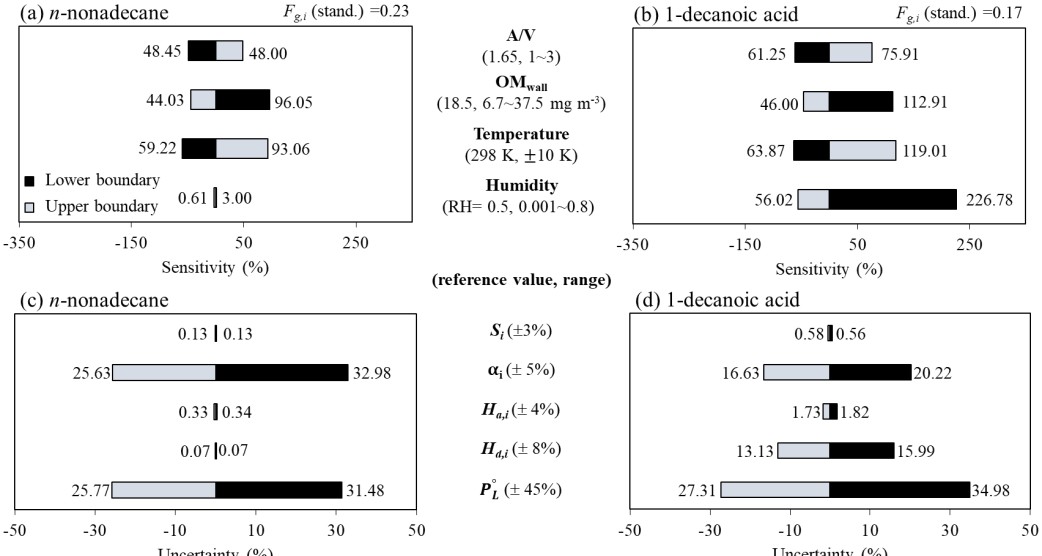

**Figure 7. The sensitivity (%) of the predicted $F_{g,i}$ (the vapor fraction of the total concentration of SVOC, Eq. (16)) to A/V, OM$_{wall}$, temperature, and humidity for *n*-nonadecane and 1-decanoic acid (a, b). The information inside the parentheses is associated with the reference value of $F_{g,i}$ and the variation of the variable. $F_{g,i}$ uncertainties (%) of the major model parameters, such as $S_i$, $\alpha_i$, $H_{d,i}$, $H_{a,i}$, and $p_L^{\circ}$ for *n*-nonadecane and 1-decanoic acid (c, d). The information inside the parentheses is associated with the range of uncertainties for each model parameter to predict $F_{g,i}$. The $F_{g,i}$ values at the standard condition for *n*-nonadecane and 1-decanoic acid are 0.23 and 0.17, respectively.**