# Peer review of "Modeling of Gas-Wall Partitioning of Organic Compounds Using a Quantitative Structure-Activity Relationship"

_Atmospheric Chemistry and Physics, 2019_

## Referee Comment (RC1) · Anonymous Referee #1 · 11 Sep 2019

General comments:

This manuscript details a modeling framework for estimating the effects of wall losses in environmental chamber experiments using a structure-activity modelling framework. The idea for this paper is creative and addresses a useful topic. Environmental chambers are a critical tool of atmospheric chemistry research and there are many chambers around the world. Not every chamber user has the equipment to accurately measure gas-phase wall losses, and such a formulation would assist in modelling of both past and future chamber experiments. Particularly, investigating the effect of humidity on GWP is useful and needed. There are, however, serious flaws in the quality of the

chamber experiments and the interpretation of those results. This work presents major discrepancies with state-of-the-art literature that are not sufficiently explained. Recently, wall loss literature has largely agreed on the principles of the phenomenon. If the authors want to disprove much of this consensus, much more rigorous examination and additional experimentation is needed.

Specific comments: There is a problem with the SVOC wall loss experiments as they are conducted in the UF-APHOR chamber. The authors inject their SVOC tracers, open the door between two chambers, mix the two chambers vigorously with a fan, and then close the door between them. This process is described as taking 10 minutes (L98 and Section S1), after which the authors start collecting SVOC on their absorbent tube at that point. Recent wall loss measurements (Zhang et al. 2015, Ye et al. 2016, Krechmer et al. 2016, 2018. Huang et al. 2018) report tau_GWP time scales from 10 to 20 minutes. Thus, the gas-wall partitioning process in the UF-APHOR chamber in this work has likely finished by the time they start measuring.

The measured tau_GWP values determined in this work (21 to 144 minutes on L 239; Table 2) are ∼1 order of magnitude longer than those reported by measurements in recent literature (Zhang et al. 2015, Ye et al. 2016, Krechmer et al. 2016, 2018. Huang et al. 2018). All report tau_GWP time scales from 10 to 20 minutes. In Krechmer et al. 2016 (Figure S1), the authors demonstrate that using a fan for active mixing significantly increases the diffusion of compounds to the walls, increasing the mixing by a factor of ∼10. Because that is what the authors do in this work, it is likely they could expect tau_gwp to be on the order of 1 minute, depending on the size of their fan. By starting measurements at 10-15 minutes after turning the fan on, the authors here have missed the bulk of the SVOC decay to the walls. Determining this mixing time scale with a trace gas such as ozone or CO2 would have been relatively simple and important for understanding these results. If the mixing time scale is < 10 minutes as expected, then starting the experiment after 10 minutes of mixing means that the vast bulk of gas-wall partitioning has occurred before measurement. The authors claim

with no experimental evidence that the UF-APHOR chamber has a longer mixing time scale. This is a major experimental weakness of this work and should be rectified before publication.

L. 99 and L312: The authors attribute their much longer tau_GWP value than other literature values to the small SA/V ratio (1.65) of the UF_APHOR chamber (vs. 3.0 in Yeh and Ziemann). According to the parameterization provided in McMurray and Grosjean [1985], the wall loss rate should actually be slightly faster than the one reported by Yeh and Ziemann. What is the reason for the wide discrepancy in modelled tau_GWP of this work?

L244 The authors also attribute the discrepancy to the high RH of the chamber. While this is possible and would be an interesting and useful result, they do not perform any experiments at the University of Florida under dry conditions. While it is useful to compare their own experiments against the Yeh and Ziemann and Matsunaga and Ziemann experments, the UC Riverside and UF chambers are different. Indeed, in other sections (L49-50), the author group here claim that the age of the chamber makes a difference in the GWP. If that were the case, then how can they use the Ziemann group results in the same model with the UF chamber results without controlling for these effects? Thus, they cannot suitably make this claim (that tau_GWP is larger due to the high RH) without additional experimental evidence.

L49-50: The model in this work assumes that gas-wall partitioning of vapors occurs by absorption into organic material (OM) deposited on the Teflon walls. This assumption has been shown previously by Matsunaga and Ziemann (2010) and Zhang et. al. (2014) to be incorrect. Matsunaga and Ziemann clearly show that gas-phase compounds are lost at equal rates and amounts to new and old chambers. Further, Matsunaga and Ziemann provide additional evidence and a mechanism based on Eyring hole theory. If the authors here want to overturn this precedent, then they need to perform experiments, such as those like Matsunaga and Ziemann with clean and dirty chamber walls and show a difference.

Technical corrections: N/A

References: Huang, Y., Zhao, R., Charan, S. M., Kenseth, C. M., Zhang, X. and Seinfeld, J. H.: Unified Theory of Vapor–Wall Mass Transport in Teflon-Walled Environmental Chambers, Environ. Sci. Technol., 52(4), 2134–2142, doi:10.1021/acs.est.7b05575, 2018.

Krechmer, J. E., Pagonis, D., Ziemann, P. J. and Jimenez, J. L. L.: Quantification of Gas-Wall Partitioning in Teflon Environmental Chambers Using Rapid Bursts of Low-Volatility Oxidized Species Generated in Situ, Environ. Sci. Technol., 50(11), 5757–5765, doi:10.1021/acs.est.6b00606, 2016.

Krechmer, J. E., Day, D. A., Ziemann, P. J. and Jimenez, J. L.: Direct Measurements of Gas/Particle Partitioning and Mass Accommodation Coefficients in Environmental Chambers, Environ. Sci. Technol., 51(20), 11867–11875, doi:10.1021/acs.est.7b02144, 2017.

Matsunaga, A. and Ziemann, P. J.: Gas-Wall Partitioning of Organic Compounds in a Teflon Film Chamber and Potential Effects on Reaction Product and Aerosol Yield Measurements, Aerosol Sci. Technol., 44(10), 881–892, doi:10.1080/02786826.2010.501044, 2010.

McMurry, P. H. and Grosjean, D.: Gas and aerosol wall losses in Teflon film smog chambers, Environ. Sci. Technol., 19(12), 1176–1182, doi:10.1021/es00142a006, 1985.

Ye, P., Ding, X., Hakala, J., Hofbauer, V., Robinson, E. S. and Donahue, N. M.: Vapor wall loss of semi-volatile organic compounds in a Teflon chamber, Aerosol Sci. Technol., 50(8), 822–834, doi:10.1080/02786826.2016.1195905, 2016.

Zhang, X., Schwantes, R. H., McVay, R. C., Lignell, H., Coggon, M. M., Flagan, R. C. and Seinfeld, J. H.: Vapor wall deposition in Teflon chambers, Atmos. Chem. Phys., 15(8), 4197–4214, doi:10.5194/acp-15-4197-2015, 2015.

Zhang, X., Cappa, C. D., Jathar, S. H., McVay, R. C., Ensberg, J. J., Kleeman, M. J.
and Seinfeld, J. H.: Influence of vapor wall loss in laboratory chambers on yields of secondary organic aerosol., Proc. Natl. Acad. Sci. U. S. A., 111(16), 5802–5807, doi:10.1073/pnas.1404727111, 2014.

---

## Referee Comment (RC2) · Anonymous Referee #3 · 12 Sep 2019

The authors report on a study that aims to allow for prediction of the loss of organic vapors to the walls of Teflon chambers based on the compound identity. This is an interesting framing of a complex problem. However, I unfortunately find that there are fundamental flaws with this study that I do not think can be addressed through revision. The two of these that I think are most important are: (i) the fundamental assumption that vapors partition only to organic matter that is bound to the walls of the teflon chamber, rather than to the chamber itself, and (ii) the quality of the data shown does not appear sufficiently high to allow for robust determination of the desired properties, and there are issues with the overall experimental design. A third issue, although one that could be addressed through revision, is that I too often found the study details

and the description of the procedures to be lacking in terms of clarity. After writing my review, I read that of Reviewer #1 and we seem to come to similar conclusions. I unfortunately do not see a path forward for this manuscript.

My specific comments follow below:

L47: I am concerned that this statement, which discusses partitioning into "absorbing organic matter on the Teflon film," demonstrates a misunderstanding of how others have been thinking about the partitioning process. There is clear evidence that the partitioning occurs into the Teflon material directly, without need for any "organic matter." Unless, of course, the authors are referring to the Teflon as "absorbing organic matter." However, by stating that the issue is partitioning into OM that is "on" the Teflon film, it makes it seem as if the authors believe the key issue to be bound OM and not the Teflon material. Adsorbed OM can contribute to the partitioning, but is not necessary. Given the discussion in Section 2.3, where the authors sample OM from the Teflon walls, it seems clear that they are considering only partitioning into wall-bound OM. But this is a small fraction of the apparent wall mass, as used by Matsunaga and Ziemann (2010), that is important to consider. This issue is also raised by Reviewer #1 and is a core limitation of the current study.

L58: it is unclear why this should be the mean molecular weight of the organic material on the Teflon. The MW term is needed to convert from vapor pressure to saturation concentration, and thus should be the MW of the SVOC under consideration.

L62: Differences in O:C ratios does not provide support for there being "diverse functionalities." If the focus is on functionalities, then I suggest the authors focus on functionalities and not generic properties such as O:C.

L68: The definition of alpha_w does not seem correct to me. It is the mass accommodation coefficient to the wall, not "the fraction that reversible uptake of a gas-phase species will occur upon collision with the chamber wall."

L96: It is not clear how the procedure the authors use "reduce[s] the delay attributable to inject chemicals." Reduces how and from what? The mixing time reported is still 10 minutes, which Krechmer et al. (2017), among others, has shown to be a very long time compared to the time to partition to walls, especially in a highly turbulent chamber as would be the case here based on the procedure described.

L104: The authors state "to ensure particle formation. . .". Was there particle formation and was this desired? This does not seem correct.

L102&108: How long were samples collected on the denuders?

L109: Based on the data presented, the measurement uncertainties are undoubtedly underestimated here. It would be useful to have a more accurate representation of the true uncertainty.

Section 2.1: The authors never directly state what chemicals they actually use. This would seem the place to give this detail.

Section 2.3: there is no mention of the RH at which the measurements are made. Are they really made at 10% and 80% RH? How was the RH controlled, if this is the case? How long were samples allowed to equilibrate? This is critical information if the statements regarding the wall-bound OM hygroscopicity are to be believed on L140.

L166: it is unclear to me how this statement is true. If the $k\_on$ increases the $k\_off$ will increase the balance and give the same $Kw$, which is an equilibrium property.

L176: is unclear why the Alpha would depend on the chamber area to volume ratio. Alpha is simply a property of the species condensing and material onto which it is condensing.

Eqn 13: It is unclear how equation 13 derives from equation 15. This needs to be shown more explicitly.

L187: I disagree with the authors Monday contend that the change in the functional

group distribution is small year around. They show a total of 2 spectra. The spectra actually show substantial differences. Further, the authors have not quantified any of the differences, taking only a very qualitative approach period if they wish to make this type of statement they need to support their conclusions more with their measurements.

Fig. 3: The data, to me, do not appear of sufficiently high quality to allow for robust determination of the model parameters derived by the authors. In many, or even most, cases the fits to the observations appear poor. I strongly suggest that a rigorous discussion of the measurement uncertainties and data quality, along with the fit quality, is required for this work to be publishable.

Section 4.3: A discussion of uncertainties is lacking entirely. The authors give no sense of whether the (for example) difference of a factor of 7 in Kw for 1-decanoic acid and n-nonadecane is statistically justified.

Overall, I find that the details provided regarding determination of the model parameters are insufficient to truly allow for this study to be reproduced. I think the authors need to do a more thorough job explaining how the calculations work and what assumptions go into them.

I do not find it clear how the authors determined the Kw values. They give a bunch of equations, but how this is determined experimentally is not clear to me. The "polynomial equation" (L179) that is used is not clear, and it is also not clear that this is really an observed value, versus a calculated value, given that the determination relies on calculated values of a variety of parameters. This, to me, makes it circular to compare the Kw to the $1/Ci^*$ values (L249). Perhaps I am simply missing the distinction, but I do not find that the authors have presented their analysis in a sufficiently clear way to understand the details.

Other Comments:

L20: I suggest that the first sentence be rewritten. As currently written, it is not a fully

formed sentence. What does "atmospheric process of reactive hydrocarbons" mean? I find this ambiguous and difficult to parse.

L23: SOA can constitute much more than 40% of the OA budget in a region. "Up to 40%" is not correct.

L26: technically, many SOA models are not mass-conserving, and thus do not take a "mass balance" approach.

L38: To what, more specifically, does "the gas-wall process" refer? Conventionally, people have used GWP (in this context) to mean gas-wall partitioning, not process.

L42: "underrated" should be "underestimated" or "under-predicted".

L44: I suggest these two sentences need be rewritten. They are very difficult to understand, yet I am also not certain that they are correct. What "database?" "Burdensome" how?

L52: The citation of Im et al. (2014) here seems unnecessary and arbitrary.

References: Krechmer, J. E., Day, D. A., Ziemann, P. J., and Jimenez, J. L.: Direct Measurements of Gas/Particle Partitioning and Mass Accommodation Coefficients in Environmental Chambers, Environmental Science & Technology, 51, 11867-11875, https://doi.org/10.1021/acs.est.7b02144, 2017.

Matsunaga, A., and Ziemann, P. J.: Gas-Wall Partitioning of Organic Compounds in a Teflon Film Chamber and Potential Effects on Reaction Product and Aerosol Yield Measurements, Aerosol Sci. Technol., 44, 881-892, https://doi.org/10.1080/02786826.2010.501044, 2010.

---

## Referee Comment (RC3) · Anonymous Referee #2 · 12 Sep 2019

Han et al. use a quantitative structure-activity relationship to predict gas-wall partitioning of semi-volatile organic compounds in chamber experiments. They explore the effects of relative humidity of gas-wall partitioning and the influences on SOA mass predictions. The approach is new and interesting. However, I have several questions and comments that needs to be addressed before I am convinced that this approach is promising to be used by other chamber users.

Major comments:

1. The elemental composition of Mwall-OM is determined as C15O4H24. It is interesting that the authors use one composition to represent the presumably tens or hundreds

of different SVOC deposited on the wall. In addition, there will be SVOC wall loss even in a completely new chamber wall without pre-deposited SOA particles and vapors. Therefore, it is not clear to me how does the Mwall-OM alone affect vapor wall loss. Further, does the wiping collect all the organic matter mass on the wall (Line 117)?

2. There is increasing evidence that secondary organic aerosols from oxidation of VOCs such as alpha-pinene consist of LVOCs and ELVOCs that contain -OOH functional groups (Bianchi et al., 2019). The authors need to broaden the discussions on the implications/limitations of using the descriptor to estimate gas-wall process and its effects on SOA mass predictions regarding -OOH and (E)LVOCs.

3. As a figure in the main text, Figure 3 deserves more description and discussions. The observed time sequences of 1-heptanoic acid and 2,5-dimethylphenol do not show a downward trend as the predicted time sequences at 40% RH. The authors need to provide more explanation.

4. Line 104: It is confusing here. Are there particles in these experiments or not? Figure 2 indicates there are no particles but line 104 indicates there are particles.

Minor comments:

1. Line 8, PaDEL-Descriptor, a software that calculates. . .

2. Table S2 and S3: There are several coefficients that have a p-value greater than 0.05, are those all included as descriptors?

Reference:

Bianchi, Federico, et al. "Highly oxygenated organic molecules (HOM) from gas-phase autoxidation involving peroxy radicals: A key contributor to atmospheric aerosol." Chemical reviews119.6 (2019): 3472-3509.
* * *

---

## Author Comment (AC1) · 18 Nov 2019

Department of Environmental Engineering Science, University of Florida, Gainesville, Florida, USA

mjang@ufl.edu

We thank reviewer 1 for the valuable comments on the manuscript.

*Overall comment:*

This manuscript details a modeling framework for estimating the effects of wall losses in environmental chamber experiments using a structure-activity modelling framework. The idea for this paper is creative and addresses a useful topic. Environmental chambers are a critical tool of atmospheric chemistry research and there are many chambers around the world. Not every chamber user has the equipment to accurately measure gas-phase wall losses, and such a formulation would assist in modelling of both past and future chamber experiments. Particularly, investigating the effect of humidity on GWP is useful and needed. There are, however, serious flaws in the quality of the chamber experiments and the interpretation of those results. This work presents major discrepancies with state-of-the-art literature that are not sufficiently explained. Recently, wall loss literature has largely agreed on the principles of the phenomenon. If the authors want to disprove much of this consensus, much more rigorous examination and additional experimentation is needed.

**Summary of response to the reviewer 1:**

Based on the valuable comments from the reviewer, this manuscript was significantly improved. In summary,

1) The chamber operation procedures (Table R1) are summarized.

2) Additional experiments

   - To verify the significance of the mixing fan on the GWP (Fig. R1).

   - To evaluate the impact of the injection and sampling time on the prediction of GWP (Fig. R2 shows the feasibility of the GWP predictive model for experiment with reduced injection and sampling time).

   - To compare the chemical composition of the organic layer ($OM_{wall}$) on the Teflon surface of the chamber wall and that of unused Teflon film by using FTIR spectra (Fig. R3)

3) Updating the coefficients in the GWP predictive polynomial equation to include the initial time (17.5 minutes) required for organic vapor transfer and organic vapor

sampling.

4) Discussion about utilizing other chamber data and GWP prediction for other chambers.

The detail responses to the comments from Reviewer 1 are following:

***Specific comments:***

***Comment 1)*** There is a problem with the SVOC wall loss experiments as they are conducted in the UF-APHOR chamber. The authors inject their SVOC tracers, open the door between two chambers, mix the two chambers vigorously with a fan, and then close the door between them. This process is described as taking 10 minutes (L98 and Section S1), after which the authors start collecting SVOC on their absorbent tube at that point. Recent wall loss measurements (Huang et al., 2018) report characteristic time for GWP ($\tau_{GWP,i}$) time scales from 10 to 20 minutes. Thus, the gas-wall partitioning process in the UF-APHOR chamber in this work has likely finished by the time they start measuring. The measured $\tau_{GWP}$ values determined in this work (21 to 144 minutes on L 239; Table. 2) are ~1 order of magnitude longer than those reported by measurements in recent literature (Zhang et al., 2015; Ye et al., 2016; Krechmer et al., 2016; Huang et al., 2018). All report $\tau_{GWP}$ time scales from 10 to 20 minutes. In Krechmer et al. (2016) (Fig. S1), the authors demonstrate that using a fan for active mixing significantly increases the diffusion of compounds to the walls, increasing the mixing by a factor of ~10. Because that is what the authors do in this work, it is likely they could expect $\tau_{GWP}$ to be on the order of 1 minute, depending on the size of their fan. By starting measurements at 10-15 minutes after turning the fan on, the authors here have missed the bulk of the SVOC decay to the walls. Determining this mixing time scale with a trace gas such as ozone or $CO_2$ would have been relatively simple and important for understanding these results. If the mixing time scale is < 10 minutes as expected, then starting the experiment after 10 minutes of mixing means that the vast bulk of gas-wall partitioning has occurred before measurement. The authors claim with no experimental evidence that the UF-APHOR chamber has a longer mixing time scale. This is a major experimental weakness of this work and should be rectified before publication.

***Response:***

In order to clarify the injection of organic vapor to the chamber, we summarized the chamber operation procedure. Table R1 is added to the supporting information.

Table R1. Chamber operation procedures.

| | Duration | Low RPM gear motor Fan | Middle door | East chamber | West chamber |
|---|---|---|---|---|---|
| Vaporization of organics to the East chamber (low RPM mixing fan on) | 30 minutes | East only | Closed | Organic vapors | Clean |
| Transfer of organic vapor from the East chamber to the West chamber | 10 minutes | Both chambers | Open | Organic vapors | Organic vapors |

| Sampling from the West chamber | Sampling began immediately after closing the door. Using the two sampling lines, the time gasp between sampling is short. | Off | Closed | No use | Use |
| --- | --- | --- | --- | --- | --- |

a.  Chemical injection time: UF-APHOR, a duel outdoor chamber (East chamber and West chamber), benefits the reduction of the time to vaporize organic species to the chamber. To avoid the chemical reaction between organic compound groups (i.e., alcohol and acid) at high temperature, the vaporization of organic compounds was performed through several batches. Generally, heating of the chemical injector under clean air streams requires at least 30 minutes for the vaporization of several batches of organic compounds. During the vaporization of organic compounds into the East chamber, the middle door (1 m x 1 m) between the two chambers was closed. After the completion of the chemical vaporization and mixing, the middle door between the two chambers was opened for 10 minutes to transfer organic vapor from the East chamber to the West chamber (see Table R1). Thus, the actual chemical injection time to the West chamber is 10 minutes. Organic vapor sampling from the West chamber began, immediately after closing the middle door,

b.  Impact of a mixing fan: In order to evaluate the impact of the mixing fan on chamber gas mixing time, we conducted the separated experiments. CCl$_4$ was introduced into the East chamber. The CCl$_4$ concentration in the East chamber monitored using GC/FID over the course of time with and without the mixing fan to observe the impact of mixing fan on the mixing time. We found that the impact of the low-RPM gear motor fan on the air mixing is negligible. As seen in the Figure R1, no difference appeared in the mixing time between mixing with the fan and that without the fan. The chamber air mixing almost completes within a time shorter than 2 minutes via the Eddie flux. The chamber is too large to be impacted by such a small fan (with a low RPM gear motor). Additionally, we measured the CCl$_4$ concentration during the transferring chemicals from the East chamber to the West chamber for 15 minutes. As shown in Fig. R1 (b), CCl$_4$ is immediately dispersed into the chamber. The mixing time is almost synchronized with the time amount for exchanging the chamber air between two chambers. Repeatedly, we found that the small mixing fan was inefficient. Thus, we conclude that the diffusion of compounds to the walls might not be affected by our mixing fan.

[Figure]

Figure R1. Impact of the low-RPM mixing fan on the relative concentration of CCl$_4$

(conc./initial conc.) in the chamber. The relative concentration of CCl₄ in the west chamber with and without mixing fan. (a) Relative concentrations of CCl₄ with no fan and with the mixing fan. (a) Relative concentrations of CCl₄ during transferring CCl₄ vapor from the East chamber to the West chamber with no fan and with the mixing fan in the West chamber.

c.  Impact of mixing time and the initial sampling time: There is an uncertainty in the calculation of the characteristic time of GWP ($k_{on}$ and $k_{off}$) associated with the time requirement for chemical injection (10 minutes) and the collection of the first sample at 7.5 minutes (mid-point of 15 minutes sampling). By considering those delays, the actual time for the first sample is 17.5 minutes. Based on this time, we updated the polynomial equations to predict $k_{on,i}$ and $K_{w,i}$. To evaluate the impact of mixing time and the sampling time on the GWP prediction, we conducted the additional experiment with the reduced air transferring time from the East to the West chamber: the first sampling time at 11.5 minutes with 8 minutes organic vapor transferring time and sampling at 3.5 minutes (7 minute sampling) after closing the door between two chambers. These data set was evaluated using the updated polynomial equations as seen in Figure R2.

[Figure]

Figure R2. The evaluation of the suitability of the updated polynomial equation against the measurement from additional chamber data (09/30/2019) with the first sampling time at 11.5 minutes (8 minutes organic vapor transferring time + sampling time at 3.5 minutes (7 minute sampling) immediately after closing the middle door between the two chambers)

***Comment 2)*** L. 99 and L312: The authors attribute their much longer characteristic time for GWP ($\tau_{GWP,i}$) value than other literature values to the small surface area/volume (A/V) ratio (1.65) of the UF-APHOR chamber (vs. 3.0 in Yeh and Ziemann). According to the parameterization provided in McMurry and Grosjean (1985), the wall loss rate should actually be slightly faster than the one reported by Yeh and Ziemann (2015). What is the reason for the wide discrepancy in modelled $\tau_{GWP,i}$ of this work?

*Response:*

It seems that the reviewer misunderstood the impact of surface/volume ratio (A/V). The smaller number means the less wall loss of organic vapor with a slower wall loss rate. Based on the Eq. 5 of McMurry and Grosjean (1985), the smaller A/V ratio leads the slower gas deposition rate ($\beta_g$).

$$-\frac{1}{C_1}\frac{dC_1}{dt} = \beta_g = \left(\frac{A}{V}\right)\frac{\alpha\bar{v}/4}{1.0 + (\pi/2)[\alpha\bar{v}/[4(k_e D)^{1/2}]]} \quad (5)$$

The characteristic time of the GWP is calculated by Eq. 13 in the manuscript.

$$\tau_{GWP} = \frac{1}{k_{on,i} + k_{off,i}}$$

The $k_{off,i}$ is related to $k_{on,i}$ and $K_{w,i}$ as follows (Eq. 7)

$$K_{w,i} = K_{OM,i}OM_{wall} = \frac{k_{on,i}}{k_{off,i}}$$

Thus, the characteristic time can be rewritten as

$$\tau_{GWP,i} = \frac{K_{W,i}}{k_{on,i}(K_{W,i}+1)}$$

The $\tau_{GWP,i}$ is generally greater with the smaller $k_{on,i}$. Thus, the $\tau_{GWP,i}$ in this study should be larger than that reported by Yeh and Ziemann (2015).

*Comment 3)* L244: The authors also attribute the discrepancy to the high RH of the chamber. While this is possible and would be an interesting and useful result, they do not perform any experiments at the University of Florida under dry conditions. While it is useful to compare their own experiments against the Yeh and Ziemann (2015) and Matsunaga and Ziemann (2010) experiments, the UC Riverside and UF chambers are different. Indeed, in other sections (L49-50), the author group here claim that the age of the chamber makes a difference in the GWP. If that were the case, then how can they use the Ziemann group results in the same model with the UF chamber results without controlling for these effects? Thus, they cannot suitably make this claim (that $\tau_{GWP,i}$ is larger due to the high RH) without additional experimental evidence.

*Response:*

a.  Limitation in the reduction of RH in the outdoor chamber: To avoid the photochemical reaction of organic vapor, all experiments were performed at night. The outdoor chamber is limited to reduce humidity at nighttime. During the daytime, humidity can reach to below 10 percent by addition of the dry tank air. However, the chamber humidity increases at nighttime when temperature drops. We waited for the stable temperature and humidity. It is difficult to reduce chamber humidity at night to the extremely dry condition (less than 0.3) because of the large volume of chamber. Prior to each experiment, the chamber air was dehumidified for three days and used two dry air tanks during the daytime at the

experimental day. Then, we can reach to 0.4 RH at night under relatively constant temperature as seen in Table 1 in the manuscript.

b. Feasibility of the QSAR approach to the data sets from different chamber: Ultimately, our predictive model aims to predict GWP of organic vapor originating from any chamber with the chamber specific parameters (OM$_{wall}$ and A/V ratio). Although the chamber specific parameters in the chamber data reported by Yeh and Ziemann (2015) and Matsunaga and Ziemann (2010) are different, we was able to use their chamber data with their chamber characteristic parameters. Additionally, the predictive GWP model was tested for two different experimental sets that were not used to the model development. Those data were different in the RH range (Figure S7 for revised SI) or the initial sampling time (Figure R2, Figure S3 was newly added to the revised manuscript).

***Comment 4)*** L49-50: The model in this work assumes that gas-wall partitioning of vapors occurs by absorption into organic material (OM) deposited on the Teflon walls. This assumption has been shown previously by Matsunaga and Ziemann (2010) and Zhang et al. (2015) to be incorrect. Matsunaga and Ziemann clearly show that gas-phase compounds are lost at equal rates and amounts to new and old chambers. Further, Matsunaga and Ziemann provide additional evidence and a mechanism based on Eyring hole theory. If the authors here want to overturn this precedent, then they need to perform experiments, such as those like Matsunaga and Ziemann with clean and dirty chamber walls and show a difference.

***Response:***

In this study, we assume that gaseous SVOCs partition onto $OM_{wall}$ of the Teflon film. $OM_{wall}$ consists of the low volatile organic mass ($M_{wall-OM}$) and the water content ($M_{wall-water}$) that is modulated by the chemical composition of $M_{wall-OM}$ and humidity. In order to characterize the chemical compositions of $OM_{wall}$, FTIR spectra of $OM_{wall}$ were measured. $OM_{wall}$ was obtained by extracting the surface of the fresh Teflon film or the aged Teflon film (the chamber used for SOA experiments) as seen in Fig. R3. The curve fitting analysis of the FTIR spectrum indicates that the aliphatic functionality (C-H stretching bend) is dominant and a small quantity of oxygenated functionalities, such as carbonyl, alcohols, and carboxylic acid, is minor in samples collected at different time and film spots. The FTIR spectrum of OM$_{wall}$ originating from the fresh Teflon film and that of the aged chamber wall shown similar spectra as seen in Fig. R3 (Fig. S4 in the revised SI).

[Figure]

Figure R3. FTIR spectra of the $OM_{wall}$ for the Teflon film chamber wall (measured on 09.30.2019 and 01.11.2019) and the unused Teflon film (measured on 10.03.2019).

The aliphatic hydrocarbons (C-H stretching bend) is major components in extracted organic matter while a small quantity of oxygenated functionality (carbonyl, alcohols, carboxylic acid, nitrate) varied in samples collected at different time. Interestingly, the adsorption of organics on the glass window has been reported in studies of indoor air (Eichler et al., 2019; Weschler and Nazaroff, 2017). Based on the study by Liu et al. (2003), the major compounds found on the glass window includes long chain alkanes and long chain-alkanoic acids, which are hydrophobic.

In particular, the fresh Teflon film has wax-like film layers. After the Teflon film is produced in industries and the film will be shipped to research institutes or chamber manufacturing companies. Most cases, the Teflon film is exposed to the ambient air and absorb wax-like materials. Furthermore, the wax-like materials slowly decay and stay on the film for many days. The Teflon film is very hydrophobic based on the Hansen's solubility parameters of Teflon (12.7) and can adsorb hydrophobic organics. Our $OM_{wall}$ composition obtained from FTIR results supports the deposition of long-chain alkanes on the chamber wall. Thus, we conclude that Teflon wall of the chamber can adsorb the organic matter from the ambient air and assume GWP as an absorption-desorption of vapor on the $OM_{wall}$. This wax-like layer would also be very viscous and may affect diffusion of SVOCs. In our model, $K_{w,i}$ can greater for the larger molecule ($\alpha_i$) suggesting that GWP is influenced by viscosity of $OM_{wall}$.

References:

Eichler, C., Cao, J., Isaacman-VanWertz, G., and Little, J.: Modeling the formation and growth of organic films on indoor surfaces, Indoor Air, 29, 17-29, 10.1111/ina.12518, 2019.
Huang, Y., Zhao, R., Charan, S., Kenseth, C., Zhang, X., and Seinfeld, J.: Unified Theory of Vapor-Wall Mass Transport in Teflon-Walled Environmental Chambers, Environmental Science & Technology, 52, 2134-2142, 10.1021/acs.est.7b05575, 2018.
Liu, Q., Chen, R., McCarry, B., Diamond, M., and Bahavar, B.: Characterization of polar organic compounds in the organic film on indoor and outdoor glass windows, Environmental

Science & Technology, 37, 2340-2349, 10.1021/es020848i, 2003.

Matsunaga, A., and Ziemann, P.: Gas-Wall Partitioning of Organic Compounds in a Teflon Film Chamber and Potential Effects on Reaction Product and Aerosol Yield Measurements, Aerosol Science and Technology, 44, 881-892, 10.1080/02786826.2010.501044, 2010.

McMurry, P., and Grosjean, D.: Gas and aerosol wall losses in Teflon film smog chambers, Environmental Science & Technology, 19, 1176-1182, 10.1021/es00142a006, 1985.

Weschler, C., and Nazaroff, W.: Growth of organic films on indoor surfaces, Indoor Air, 27, 1101-1112, 10.1111/ina.12396, 2017.

Yeh, G., and Ziemann, P.: Gas-Wall Partitioning of Oxygenated Organic Compounds: Measurements, Structure-Activity Relationships, and Correlation with Gas Chromatographic Retention Factor, Aerosol Science and Technology, 49, 726-737, 10.1080/02786826.2015.1068427, 2015.

---

## Author Comment (AC2) · 18 Nov 2019

Department of Environmental Engineering Science, University of Florida, Gainesville, Florida, USA

mjang@ufl.edu

We thank reviewer 3 for the valuable comments on the manuscript.

*Overall comment:*

The authors report on a study that aims to allow for prediction of the loss of organic vapors to the walls of Teflon chambers based on the compound identity. This is an interesting framing of a complex problem. However, I unfortunately find that there are fundamental flaws with this study that I do not think can be addressed through revision. The two of these that I think are most important are: (i) the fundamental assumption that vapors partition only to organic matter that is bound to the walls of the Teflon chamber, rather than to the chamber itself, and (ii) the quality of the data shown does not appear sufficiently high to allow for robust determination of the desired properties, and there are issues with the overall experimental design. A third issue, although one that could be addressed through revision, is that I too often found the study details and the description of the procedures to be lacking in terms of clarity. After writing my review, I read that of Reviewer #1 and we seem to come to similar conclusions. I unfortunately do not see a path forward for this manuscript.

I find that the details provided regarding determination of the model parameters are insufficient to truly allow for this study to be reproduced. I think the authors need to do a more thorough job explaining how the calculations work and what assumptions go into them. I do not find it clear how the authors determined the $K_{w,i}$ values. They give a bunch of equations, but how this is determined experimentally is not clear to me. The "polynomial equation" (L179) that is used is not clear, and it is also not clear that this is really an observed value, versus a calculated value, given that the determination relies on calculated values of a variety of parameters. This, to me, makes it circular to compare the $K_{w,i}$ to the $1/C_i^*$ values (L249). Perhaps I am simply missing the distinction, but I do not find that the authors have presented their analysis in a sufficiently clear way to understand the details.

*Summary of Response to Reviewer 3:*

Based on the valuable comments from the reviewer, this manuscript was significantly improved. In summary

1) The gas-wall partitioning (GWP) of SVOCs is assumed as absorption-desorption kinetics to organic layer on the Teflon surface ($OM_{wall}$). This assumption set based on our

measurement of extracted $OM_{wall}$. The estimated composition and water contents of $OM_{wall}$ using FTIR spectrum supports that the $OM_{wall}$ is wax-like hydrophobic compounds (response to comment 1) which can dominantly impact on the GWP.

2) Model uncertainty

The uncertainties of predicted $K_{w,i}$ and $\alpha_{w,i}$ for each species were calculated based on the 95% confidence level boundary of each coefficient and the uncertainties of each physicochemial parameter in the polynomial equation to predict $K_{w,i}$ and $\alpha_{w,i}$. The uncertainties were presented with the predicted value on Table R2. Table 2 is updated with uncertainties in the revised manuscript.

3) Data uncertainty

The uncertainties of experimental data points in Fig. 3 of the revised manuscript were determined with a propagation error based on the quantitative procedure. The error associated with $C_{g,i}$ ranges 10-30% .

4) Procedure to estimate $OM_{wall}$ composition

To response to the comment from the reviewer, the additional characterization of the organic layer ($OM_{wall}$) on the Teflon film surface of the chamber wall and the unused Teflon film was performed by using FTIR spectra.

The detail responses to the comments from Reviewer 3 are following:

***Specific comments:***

***Comment 1)*** L47: I am concerned that this statement, which discusses partitioning into "absorbing organic matter on the Teflon film," demonstrates a misunderstanding of how others have been thinking about the partitioning process. There is clear evidence that the partitioning occurs into the Teflon material directly, without need for any "organic matter." Unless, of course, the authors are referring to the Teflon as "absorbing organic matter." However, by stating that the issue is partitioning into OM that is "on" the Teflon film, it makes it seem as if the authors believe the key issue to be bound OM and not the Teflon material. Adsorbed OM can contribute to the partitioning, but is not necessary. Given the discussion in Section 2.3, where the authors sample OM from the Teflon walls, it seems clear that they are considering only partitioning into wall-bound OM. But this is a small fraction of the apparent wall mass, as used by Matsunaga and Ziemann (2010), that is important to consider. This issue is also raised by Reviewer #1 and is a core limitation of the current study.

***Response:***

The quantity and chemical properties of $OM_{wall}$ varied between laboratories as summarized in Table R1 below:

Table R1. Summarized the assumption of the GWP on several previous studies.

| Publication | Assumption | Chemical property of the surface layer | MW of wall material (g/mol) | chamber style (indoor/outdoor) |
|---|---|---|---|---|
| (Matsunaga and Ziemann, 2010;Krechmer et al., 2016) | Sorption mechanism (Eyring Hole theory) of SVOCs to the Teflon surface layer with effective mass concentration of organic aerosol particles ($C_w$) [a]. | Teflon surface with $C_w$ (absorbed SOA products) | 200/250 | Indoor |
| (Huang et al., 2018) | Two layers consisting of the surface Teflon layer and inner Teflon layer. | Surface Teflon layer with $C_w$ (absorbed SOA products) | 200 | Indoor |
| This study | Absorption of SVOCs to the viscous organic layer on the Teflon film wall. | Wax-like organic matter | 273 | Outdoor |

[a] The equivalent mass concentration ($C_w$) reported by Krechmer et al. (2016) is described as the absorbed SOA products on the Teflon surface by usage of the chamber. It is theoretically calculated based on the chamber history.

In this study, we assumed that organic vapor partition on the viscous organic layer on the Teflon film. This assumption was established based on our measurement of the chemical property of the solvent-extractable organic matter on the Teflon film wall. As seen the response to comment 4 from reviewer 1, the hydrophobic Teflon film favors the adsorption of wax-like organics in ambient air. Wax-like carbons were observed even in unused Teflon film by using FTIR. With the assumption of homogeneous coating of the Teflon film with $OM_{wall}$, the estimated thickness of the wax-like materials ranges 7.5-10.0 nm (calculated with the extracted OM mass (~800 μg) from the film surface area (~1200 cm$^2$)). This thickness suggests multi-layer coating. The organic layer suggested by Krechmer et al. (2016) was a monolayer (1.5 nm or smaller) based on their theoretical assumption.

Based on our FTIR study (Fig. R3 in the response to comment 4 from reviewer 1), the estimated molecular weight of $OM_{wall}$ is about 273 g/mol. The hygroscopicity of the Teflon film (Fig. S3), which is measured using FTIR data, suggests that the film surface material is hydrophobic. FTIR data shows that the water content associated with the film dramatically changes at the RH higher than 0.7. This viscous and hydrophobic $OM_{wall}$ layer can impact on GWP. Shiraiwa et al. (2011) reported that the diffusivity of atmospheric chemical species in semisolid aerosol materials is very slow with a time scale ranging from several hours to days. The wax-like material determined by our study may also penetrate the certain layers of the Teflon film near the surface and influence the property of the Teflon film surface.

In the current knowledge, the understanding of the characteristics of $OM_{wall}$ is uncertain. The molecular weight of $OM_{wall}$ in the previous studies was determined by using the average molecular weight of SOA based on the value in the literature. For example, Matsunaga and Ziemann (2010) and Krechmer et al. (2016) determined the $OM_{wall}$'s molecular weight based on SOA studies reported by Seinfeld et al. (2001). In their studies, the $OM_{wall}$ molecular weight was not determined theoretically or by measuring of the chemical composition of Teflon surface matter. In the study by Krechmer et al. (2016), the $OM_{wall}$ molecular weight was 250 g/mol, which is the same molecular weight with a FEP monomer. However, the actual molecular weight of FEP ranges 76,000-603,000 (Wypych, 2016). Furthermore, the monomeric

FEP with 250 g/mol molecular weight is very volatile (order of $10^3$ mmHg) because Teflon functionality significantly reduces volatility. Thus, the monomeric FEP are gaseous and it cannot be regarded as the molecular weight of a surface material. Thus, the proposed molecular weight (200-250 g/mol) in the previous studies are uncertain. Importantly, either wax-like matter of this study or the short chain-length Teflon layer used in other studies are viscous and they will slow down the diffusion of organic molecules. Our QSAR approach to produce predictive polynomial equations for GWP is developed based on the physicochemical properties of organic compounds and will not be influenced by any assumption for $OM_{wall}$ at given A/V and $MW_{OM}$.

***Comment 2)*** L58: it is unclear why this should be the mean molecular weight of the organic material on the Teflon. The MW term is needed to convert from vapor pressure to saturation concentration, and thus should be the MW of the SVOC under consideration.

***Response:***

Please find the response to comment 1.

Based on our determined composition, the vapor pressure of an organic compound is in the order of $10^{-9}$ mmHg, suggesting that this compound is nearly nonvolatile.

***Comment 3)*** L62: Differences in O:C ratios does not provide support for there being "diverse functionalities." If the focus is on functionalities, then I suggest the authors focus on functionalities and not generic properties such as O:C.

***Response:***

The sentence pointed by the reviewer has been revised and reads now,

"SOA products originating from the oxidation of hydrocarbons are diverse in functionalities, such as alkanes, aldehyde, carboxylic acids, ketones, and alcohols. The polarity of SOA is closely related to the oxygen to carbon (O:C) ratio of SOA. For example, the O:C ratio of α-pinene SOA is 0.43 on average (Zhang et al., 2015; Chen et al., 2011) but that of isoprene products is about 0.8 (Bertram et al., 2011; Chen et al., 2011; Kuwata et al., 2013)."

***Comment 4)*** L68: The definition of $\alpha_{w,i}$ does not seem correct to me. It is the mass accommodation coefficient to the wall, not "the fraction that reversible uptake of a gas-phase species will occur upon collision with the chamber wall."

***Response:***

This sentence has been removed

***Comment 5)*** L96: It is not clear how the procedure the authors use "reduce[s] the delay attributable to inject chemicals." Reduces how and from what? The mixing time reported is still 10 minutes, which Krechmer et al. (2016), among others, has shown to be a very long time compared to the time to partition to walls, especially in a highly turbulent chamber as would be the case here based on the procedure described.

*Response:*

The summary table for the injection of organic vapor into the chamber is shown in the response to comment 1 from reviewer 1 (Please find the Table R1 in the response to comment 1 from reviewer 1). In order to avoid the potential condensation reactions between different chemical functionalities, each class of organic compounds was separately vaporized into the chamber. With the three groups, the injection of 21 compounds into the East chamber takes at least 30 minutes. UF-APHOR is a duel chamber system with the air tighten door in the middle. The organic vapor introduced into the East chamber was transferred to the West for 10 minutes by opening the middle door.

*Comment 6)* L104: The authors state "to ensure particle formation. . .". Was there particle formation and was this desired? This does not seem correct.

*Response:*

Thank you for pointing this sentence. This has been corrected to provide our original intention to readers and reads now,

"No particle appeared after vaporizing organic chemicals into the chamber air based on particle data, which was monitored using a scanning mobility particle sizer (SMPS, TSI 3080, Shoreview, MN, USA) and a condensation particle counter (CPC, TSI 3022A, Shoreview, MN, USA)."

*Comment 7)* L102&108: How long were samples collected on the denuders?

*Response:*

It is mentioned in the SI section 1. The sampling duration for the first sample was 15 minutes and was gradually increased up to 35-40 minutes.

*Comment 8)* L109: Based on the data presented, the measurement uncertainties are undoubtedly underestimated here. It would be useful to have a more accurate representation of the true uncertainty.

*Response:*

The information of the error associated with the organic vapor concentration shown in Fig. 3 was updated and included in the figure caption. The concentration of gas phase SVOCs are calculated by the equation below:

$$C_{g,i} = \frac{f\left(\frac{peak\ area\ of\ target\ species}{peak\ area\ of\ internal\ standard}\right) \times dilution\ factor}{Flow\ rate \times sampling\ time}$$

*f(x)* is the calibration curve for each compound obtained from external standard. The error was estimated with a propagation error calculation method. Thus, the calculated uncertainty of data points includes flow rate, sampling time, GC measurement, and the uncertainty of the GC calibration curve. The calculated uncertainty ranges 10-30% of the concentration and it is

updated on the Fig. 3 and the caption of Fig. 3 as follows:

[Figure]

**Figure 3. The observed concentrations of the gaseous SVOCs ($C_{g,i}$, plot) in the chamber air for the two experiments at two different humidity (October 8, 2018 at RH=0.40 and May 15, 2018 at RH= 0.75) and predicted $C_{g,i}$ (line) as a time series. $C_{g,i}$ was calculated by the equation ( $C_{g,i} = \dfrac{f\left(\frac{peak\ area\ of\ target\ species}{peak\ area\ of\ internal\ standard}\right)\times dilution\ factor}{Flow\ rate\times sampling\ time}$ ). The error bar associate with $C_{g,i}$ estimated using propagation error based on the quantitating procedure of GC/MS data ranges 10–30% of the estimated concentrations.**

*Comment 9)* Section 2.1: The authors never directly state what chemicals they actually use. This would seem the place to give this detail.

*Response:*

Table 1 in the main manuscript and the section S1 in supporting materials provide the chemical information used in this study.

*Comment 10)* Section 2.3: there is no mention of the RH at which the measurements are made. Are they really made at 10% and 80% RH? How was the RH controlled, if this is the case? How long were samples allowed to equilibrate? This is critical information if the statements regarding the wall-bound OM hygroscopicity are to be believed on L140.

*Response:*

The detail information of the experimental procedure is described in the Hygroscopicity of OM$_{wall}$ in Section S2. In order to respond to the reviewer, we added the information about humidity change rate into Section S2 in the revised SI.

"Relative humidity increased to 0.8, stands for 15 minutes, and decreased at 0.01/minutes. Hygroscopicity experiments using FTIR were repeated three times."

*Comment 11)* L166: it is unclear to me how this statement is true. If the $k_{on,i}$ increases the $k_{off,i}$ will increase the balance and give the same $K_{w,i}$, which is an equilibrium property.

*Response:*

This has been corrected.

"The larger A/V ratio can yield the larger $k_{on,i}$, and thus $k_{off,i}$, at a given $K_{w,i}$ (Eq. 7), ultimately leading the shorter GWP equilibrium time (see Sect. 3.2)."

***Comment 12)*** L176: is unclear why the $\alpha_{w,i}$ would depend on the chamber area to volume ratio. Alpha is simply a property of the species condensing and material onto which it is condensing.

***Response:***

This has been corrected and reads now,

"The prediction of $k_{on,i}$, which is dependent on $\alpha_{w,i}$ and the chamber's characteristics (A/V).

The $\alpha_{w,i}$ diverged with the physicochemical properties of SVOC can also be predicted by using

QSAR approach as follows:"

***Comment 13)*** Eq. 13: It is unclear how Eq. 13 derives from Eq. 5. This needs to be shown more explicitly.

***Response:***

The coefficient in the exponential term in Eq. 5 is a rate constant consisting of $k_{on,i}$ and $k_{off,i}$. The characteristic time is inversely related to the rate constant.

***Comment 14)*** L187: I disagree with the authors contend that the change in the functional group distribution is small year around. They show a total of 2 spectra. The spectra actually show substantial differences. Further, the authors have not quantified any of the differences, taking only a very qualitative approach period if they wish to make this type of statement they need to support their conclusions more with their measurements.

***Response:***

a.  Difference between two FTIR spectra: Based on the chamber log, high concentration of inorganic seed was injected before the *OM_wall* extraction on 03/14/2019. The FTIR spectrum measured on 03/14/2019 might affected by inorganic seed. Thus, additional extracted OM_wall from chamber have been analyzed and shown in Fig. R3 for the response to comment 4 from Reviewer 1. Fig. S4 is replaced with Fig. R3 for the response to comment 4 from Reviewer 1.

b.  Quantification of the FTIR spectrum: In order to response to the reviewer, Fig. 1 and Fig. S4 are updated the detail information was added below Table S1.

"The procedure of least-squares curve fitting was implemented for the multi-component analysis. The fitting parameters include the center frequency, the peak absorbance, and the half width at half-height. The band shapes in the FTIR absorbance spectrum were approximated by a Gaussian function (Li et al., 2016;Jang and Kamens, 2001;Jang et al., 2008). The decoupled the FTIR bend for each functional group was applied to estimate the functionality composition of OM_wall using the relative intensity of the functional group determined from various reference compounds."

***Comment 15)*** Fig. 3: The data, to me, do not appear of sufficiently high quality to allow for

robust determination of the model parameters derived by the authors. In many, or even most, cases the fits to the observations appear poor. I strongly suggest that a rigorous discussion of the measurement uncertainties and data quality, along with the fit quality, is required for this work to be publishable.

***Response:***

Please find the response to comment 8.

The uncertainty of experimentally measured $C_{g,i}$ was updated on Fig. 3 and the information of the uncertainty of measurement data was updated on the figure caption.

***Comment 16)*** Section 4.3: A discussion of uncertainties is lacking entirely. The authors give no sense of whether the (for example) difference of a factor of 7 in $K_{w,i}$ for 1-decanoic acid and n-nonadecane is statistically justified.

***Response:***

In order to response to the reviewer, the model uncertainty analysis was performed for $K_{w,i}$ and $\alpha_{w,i}$. The uncertainties of predicted $K_{w,i}$ and $\alpha_{w,i}$ were determined from the standard error associating with the final polynomial equations (Eq. 14 and Eq. 15, respectively) and uncertainty of each physicochemical descriptor.

The discussion in the Section 4.3 has been revised based on the updated polynomial equation and uncertainty analysis and reads now,

"For example, 1-decanoic acid's $K_{w,i}$ is nearly 4 times higher ($18.72 \pm 6.6$) than $n$-nonadecane's $K_{w,i}$ ($5.34 \pm 2.5$), although 1-decanoic acid's estimated $p_{L,i}^{\circ}$ ($1.18 \times 10^{-3}$ mmHg) is close to $n$-nonadecane's $p_{L,i}^{\circ}$ ($1.09 \times 10^{-3}$ mmHg). In general, a longer $\tau_{GWP}$ was found for SVOCs with high $K_{w,i}$ and low $\alpha_{w,i}$. For example, $n$-eicosane's $\tau_{GWP}$ (127 min) was significantly higher than that of 2-heptanol (30 min), while $n$-eicosane and 2-heptanol had a similar $\alpha_{w,i}$ ($2.9 (\pm 0.2) \times 10^{-6}$ and $3.3 (\pm 0.1) \times 10^{-6}$, respectively). The dissimilar $K_{w,i}$ values of $n$-eicosane and 2-heptanol ($7.49 \pm 3.5$ and $0.44 \pm 0.2$, respectively) occurred with large differences in $\tau_{GWP}$."

In addition, Table 2 in the manuscript is updated with uncertainties.

**Table R2. Chemicals injected into the chamber and their physicochemical parameters (at 298 K) predicted using PaDel-Descriptor and estimated $K_{w,i}$, $\alpha_{w,i}$, and $\tau_{GWP}$ under a given condition (298K, RH=0.75).**

| No. | Chemical | MW | Molecular descriptors [a] | | | | | $p_L^{\circ}$ [b] | $K_{w,i} \pm unc$ [c] | $\alpha_{w,i} \pm unc$ [c] ($\times 10^{-6}$) | $\tau_{GWP}$ (s) |
|---|---|---|---|---|---|---|---|---|---|---|---|
| | | | $H_{d,i}$ | $H_{a,i}$ | $E_i$ | $S_i$ | $\alpha_i$ | | | | |
| 1 | 1-hexanoic acid | 116 | 0.59 | 0.41 | 0.15 | 0.51 | 14.08 | 0.078 | 1.27$\pm$0.5 | 2.7$\pm$0.2 | 3671 |
| 2 | 1-heptanoic acid | 130 | 0.59 | 0.41 | 0.15 | 0.51 | 16.26 | 0.026 | 2.64$\pm$1.0 | 2.5$\pm$0.1 | 5185 |
| 3 | 1-octanoic acid | 144 | 0.59 | 0.41 | 0.15 | 0.51 | 18.45 | 0.009 | 5.33$\pm$2.0 | 2.3$\pm$0.1 | 6563 |
| 4 | 1-decanoic acid | 172 | 0.59 | 0.41 | 0.15 | 0.51 | 22.82 | 0.001 | 18.72$\pm$6.6 | 1.9$\pm$0.1 | 8832 |
| 5 | Benzoic acid | 122 | 0.59 | 0.46 | 0.78 | 0.93 | 7.52 | 0.020 | 13.39$\pm$4.9 | 2.2$\pm$0.1 | 7097 |
| 6 | Pyruvic acid | 88 | 0.43 | 0.68 | 0.29 | 0.88 | 8.48 | 0.256 | 1.69$\pm$0.6 | 1.9$\pm$0.1 | 4661 |
| 7 | 2-heptanol | 116 | 0.35 | 0.40 | 0.19 | 0.41 | 17.49 | 0.317 | 0.44$\pm$0.2 | 3.3$\pm$0.1 | 1799 |
| 8 | 1-octanol | 130 | 0.35 | 0.39 | 0.21 | 0.45 | 19.68 | 0.109 | 0.86$\pm$0.4 | 3.0$\pm$0.2 | 2975 |
| 9 | 1-nonanol | 144 | 0.35 | 0.39 | 0.21 | 0.45 | 21.86 | 0.036 | 1.82$\pm$0.8 | 2.7$\pm$0.2 | 4499 |
| 10 | Benzyl alcohol | 108 | 0.35 | 0.56 | 0.83 | 0.88 | 8.75 | 0.093 | 5.85$\pm$2.4 | 2.4$\pm$0.1 | 5904 |
| 11 | Phenol | 94 | 0.55 | 0.43 | 0.83 | 0.88 | 6.56 | 0.988 | 0.38$\pm$0.2 | 2.6$\pm$0.2 | 1735 |
| 12 | 2,5-dimenthylphenol | 122 | 0.55 | 0.43 | 0.85 | 0.83 | 10.93 | 0.123 | 1.44$\pm$0.6 | 2.3$\pm$0.1 | 4352 |
| 13 | 2,6-dimethoxyphenol | 154 | 0.04 | 0.284 | 0.74 | 1.36 | 14.76 | 0.041 | 11.45$\pm$6.0 | 2.2$\pm$0.1 | 7458 |
| 14 | n-heptadecane | 240 | 0.00 | 0.08 | 0.04 | 0.13 | 39.36 | 0.005 | 2.43$\pm$1.1 | 3.8$\pm$0.3 | 4726 |
| 15 | n-nonadecane | 308 | 0.00 | 0.08 | 0.04 | 0.13 | 43.73 | 0.001 | 5.34$\pm$2.5 | 3.2$\pm$0.2 | 6758 |
| 16 | n-eicosane | 324 | 0.00 | 0.08 | 0.04 | 0.13 | 45.91 | 0.001 | 7.49$\pm$3.5 | 2.9$\pm$0.2 | 7629 |
| 17 | 2-Dodecanone | 184 | 0.00 | 0.42 | 0.18 | 0.60 | 27.19 | 0.053 | 1.69$\pm$0.8 | 2.5$\pm$0.1 | 4982 |
| 18 | 2-Tridecanone | 198 | 0.00 | 0.42 | 0.18 | 0.60 | 29.38 | 0.022 | 2.74$\pm$1.3 | 2.3$\pm$0.1 | 6314 |
| 19 | Decanal | 156 | 0.00 | 0.42 | 0.15 | 0.51 | 22.82 | 0.193 | 0.97$\pm$0.5 | 2.9$\pm$0.2 | 3390 |
| 20 | Citral | 152 | 0.00 | 0.46 | 0.15 | 0.51 | 18.45 | 0.142 | 2.66$\pm$1.3 | 2.8$\pm$0.2 | 5032 |
| 21 | Benzaldehyde | 106 | 0.00 | 0.47 | 0.15 | 0.51 | 7.52 | 0.982 | 2.26$\pm$1.1 | 3.3$\pm$0.2 | 3995 |

[a] Based on the QSAR approach with PaDEL-Descriptor (Yap, 2011).

[b] Calculated through group contribution (Zhao et al., 1999;Stein and Brown, 1994).

[c] The uncertainty propagation error calculated based on the 95% confidence level boundary of predictive polynomial equations for $K_{w,i}$ and $\alpha_{w,i}$.

***Comment 17)*** L20: I suggest that the first sentence be rewritten. As currently written, it is not a fully formed sentence. What does "atmospheric process of reactive hydrocarbons" mean? I find this ambiguous and difficult to parse.

***Response:***

The sentence pointed by the reviewer has been revised and reads now,

"Organic Aerosol (OA) consists of primary sources originating from fuel combustion, industries, and vehicles and Secondary Organic Aerosol (SOA) which forms via the atmospheric reactions of reactive hydrocarbons with atmospheric oxidants (i.e., an OH radical, ozone, and a nitrate radical)."

***Comment 18)*** L23: SOA can constitute much more than 40% of the OA budget in a region. "Up to 40%" is not correct.

***Response:***

The sentence pointed by the reviewer has been revised and reads now,

 "SOA constitutes a large proportion (ranging from 20% to 90%) of OA in the ambient air (Hallquist et al., 2009) and it can significantly impact on climate (Seinfeld and Pandis, 2016), visibility (Park et al., 2003), and human health (Cohen et al., 2017)."

***Comment 19)*** L26: technically, many SOA models are not mass-conserving, and thus do not take a "mass balance" approach.

***Response:***

This sentence has been removed.

***Comment 20)*** L38: To what, more specifically, does "the gas-wall process" refer? Conventionally, people have used GWP (in this context) to mean gas-wall partitioning, not process.

***Response:***

The definition of GWP has been changed from the gas-wall process to gas-wall partitioning.

***Comment 21)*** L42: "underrated" should be "underestimated" or "under-predicted".

***Response:***

This has been corrected and reads now,

"La et al. (2016) indicated that the SOA yield inferred from a chamber study can be under-predicted by more than 50% for alkane and alkene series."

***Comment 22)*** L44: I suggest these two sentences need be rewritten. They are very difficult to understand, yet I am also not certain that they are correct. What "database?" "Burdensome" how?

***Response:***

The sentence pointed by the reviewer has been revised and reads now,

"To date, the predictive model to account for the impact of GWP on SOA yields remains controversial due to the limitations of the experimental approach to measures the vapor concentration of organic species. For example, the actual vapor concentration of the organic compound in the initial point is burdensome because the experimental procedure required for the vaporization of organic species into a chamber delays the measurement of the initial concentration of organic vapor without GWP."

***Comment 23)*** L52: The citation of Im et al. (2014) here seems unnecessary and arbitrary.

***Response:***
This has been removed.

References:

Bertram, A., Martin, S., Hanna, S., Smith, M., Bodsworth, A., Chen, Q., Kuwata, M., Liu, A., You, Y., and Zorn, S.: Predicting the relative humidities of liquid-liquid phase separation, efflorescence, and deliquescence of mixed particles of ammonium sulfate, organic material, and water using the organic-to-sulfate mass ratio of the particle and the oxygen-to-carbon elemental ratio of the organic component, Atmospheric Chemistry and Physics, 11, 10995-11006, 2011.
Chen, Q., Liu, Y., Donahue, N. M., Shilling, J. E., and Martin, S. T.: Particle-phase chemistry of secondary organic material: modeled compared to measured O: C and H: C elemental ratios provide constraints, Environmental science & technology, 45, 4763-4770, 2011.
Cohen, A., Brauer, M., Burnett, R., Anderson, H., Frostad, J., Estep, K., Balakrishnan, K., Brunekreef, B., Dandona, L., Dandona, R., Feigin, V., Freedman, G., Hubbell, B., Jobling, A., Kan, H., Knibbs, L., Liu, Y., Martin, R., Morawska, L., Pope, C., Shin, H., Straif, K., Shaddick, G., Thomas, M., van Dingenen, R., van Donkelaar, A., Vos, T., Murray, C., and Forouzanfar, M.: Estimates and 25-year trends of the global burden of disease attributable to ambient air pollution: an analysis of data from the Global Burden of Diseases Study 2015, Lancet, 389, 1907-1918, 10.1016/S0140-6736(17)30505-6, 2017.
Hallquist, M., Wenger, J., Baltensperger, U., Rudich, Y., Simpson, D., Claeys, M., Dommen, J., Donahue, N., George, C., Goldstein, A., Hamilton, J., Herrmann, H., Hoffmann, T., Iinuma, Y., Jang, M., Jenkin, M., Jimenez, J., Kiendler-Scharr, A., Maenhaut, W., McFiggans, G., Mentel, T., Monod, A., Prevot, A., Seinfeld, J., Surratt, J., Szmigielski, R., and Wildt, J.: The formation, properties and impact of secondary organic aerosol: current and emerging issues, Atmospheric Chemistry and Physics, 9, 5155-5236, 10.5194/acp-9-5155-2009, 2009.
Huang, Y., Zhao, R., Charan, S., Kenseth, C., Zhang, X., and Seinfeld, J.: Unified Theory of Vapor-Wall Mass Transport in Teflon-Walled Environmental Chambers, Environmental Science & Technology, 52, 2134-2142, 10.1021/acs.est.7b05575, 2018.
Jang, M., and Kamens, R.: Atmospheric secondary aerosol formation by heterogeneous reactions of aldehydes in the presence of a sulfuric acid aerosol catalyst, Environmental Science & Technology, 35, 4758-4766, 10.1021/es010790s, 2001.
Jang, M., Cao, G., and Paul, J.: Colorimetric particle acidity analysis of secondary organic aerosol coating on submicron acidic aerosols, Aerosol Science and Technology, 42, 409-420,

10.1080/02786820802154861, 2008.

Krechmer, J., Pagonis, D., Ziemann, P., and Jimenez, J.: Quantification of Gas-Wall Partitioning in Teflon Environmental Chambers Using Rapid Bursts of Low-Volatility Oxidized Species Generated in Situ, Environmental Science & Technology, 50, 5757-5765, 10.1021/acs.est.6b00606, 2016.

Kuwata, M., Shao, W., Lebouteiller, R., and Martin, S.: Classifying organic materials by oxygen-to-carbon elemental ratio to predict the activation regime of Cloud Condensation Nuclei (CCN), Atmospheric Chemistry and Physics, 13, 5309-5324, 2013.

Li, J., Jang, M., and Beardsley, R.: Dialkylsulfate formation in sulfuric acid-seeded secondary organic aerosol produced using an outdoor chamber under natural sunlight, Environmental Chemistry, 13, 590-601, 10.1071/EN15129, 2016.

Matsunaga, A., and Ziemann, P.: Gas-Wall Partitioning of Organic Compounds in a Teflon Film Chamber and Potential Effects on Reaction Product and Aerosol Yield Measurements, Aerosol Science and Technology, 44, 881-892, 10.1080/02786826.2010.501044, 2010.

Park, R., Jacob, D., Chin, M., and Martin, R.: Sources of carbonaceous aerosols over the United States and implications for natural visibility, Journal of Geophysical Research-Atmospheres, 108, 10.1029/2002JD003190, 2003.

Seinfeld, J., Erdakos, G., Asher, W., and Pankow, J.: Modeling the formation of secondary organic aerosol (SOA). 2. The predicted effects of relative humidity on aerosol formation in the alpha-pinene-, beta-pinene-, sabinene-, Delta(3)-Carene-, and cyclohexene-ozone systems, Environmental Science & Technology, 35, 1806-1817, 10.1021/es001765+, 2001.

Seinfeld, J. H., and Pandis, S. N.: Atmospheric chemistry and physics: from air pollution to climate change, John Wiley & Sons, 2016.

Shiraiwa, M., Ammann, M., Koop, T., and Poschl, U.: Gas uptake and chemical aging of semisolid organic aerosol particles, Proceedings of the National Academy of Sciences of the United States of America, 108, 11003-11008, 10.1073/pnas.1103045108, 2011.

Volkamer, R., Jimenez, J., San Martini, F., Dzepina, K., Zhang, Q., Salcedo, D., Molina, L., Worsnop, D., and Molina, M.: Secondary organic aerosol formation from anthropogenic air pollution: Rapid and higher than expected, Geophysical Research Letters, 33, 10.1029/2006GL026899, 2006.

Wypych, G.: Handbook of polymers, Elsevier, 2016.

Zhang, X., McVay, R. C., Huang, D. D., Dalleska, N. F., Aumont, B., Flagan, R. C., and Seinfeld, J. H.: Formation and evolution of molecular products in alpha-pinene secondary organic aerosol, Proc Natl Acad Sci U S A, 112, 14168-14173, 10.1073/pnas.1517742112, 2015.

---

## Author Comment (AC3) · 18 Nov 2019

Department of Environmental Engineering Science, University of Florida, Gainesville, Florida, USA

mjang@ufl.edu

We thank reviewer 2 for the valuable comments on the manuscript.

*Overall comment:*

Han et al. use a quantitative structure-activity relationship to predict gas-wall partitioning of semi-volatile organic compounds in chamber experiments. They explore the effects of relative humidity of gas-wall partitioning and the influences on SOA mass predictions. The approach is new and interesting. However, I have several questions and comments that needs to be addressed before I am convinced that this approach is promising to be used by other chamber users.

*Summary of response to the reviewer 2:*

1) Through additional experiments, we characterize the chemical composition of the organic layer ($OM_{wall}$) on the surface of both the Teflon film chamber wall and the unused Teflon film by using FTIR spectra (Fig. R3, please find the responds to comment 4 from reviewer 1)

2) The limitation of the extraction method to characterize of $OM_{wall}$ was discussed (response b to comment 1).

3) The suitability of the QSAR-base GWP model to other products from oxidation of reactive hydrocarbons was discussed.

4) The GWP model was derived using the actual sampling time by including the duration to introduce organic vapor into the West chamber and the sampling duration (total 17.5 minutes). To demonstrate the feasibility of the GWP model, we conducted the additional chamber experiment with the short time lag (12.5 minutes) from time =0.

5) The detail information about the determination of the descriptor for the GWP predictive polynomial equation was discussed and this was added to the revised SI.

The detail responses to the comments from Reviewer 2 are following:

*Major comments:*

*Comment 1)* The elemental composition of $M_{wall\text{-}OM}$ is determined as $C_{15}H_{24}O_4$. It is interesting that the authors use one composition to represent the presumably tens or hundreds of different SVOC deposited on the wall. In addition, there will be SVOC wall loss even in a completely new chamber wall without pre-deposited SOA particles and vapors. Therefore, it is not clear to

me how does the $M_{wall-OM}$ alone affect vapor wall loss. Further, does the wiping collect all the organic matter mass on the wall (Line 117)?

***Response:***

Please also find the response to comment 4 from reviewer 1 and the response to comment 1 from reviewer 3.

    a. *Composition of OM on wall:* In order to response the reviewer, we measured FTIR spectra of the organic matter collected from the Teflon film wall inside the UF-APHOR chamber at different time. We also included the FTIR spectrum of organic mass originating from the unused Teflon film as seen in Fig. R3 for response to reviewer 1. Regardless of aging time associated with chamber history, a strong aliphatic carbon peak (wax-like material) appeared in all spectra. Teflon would be the most hydrophobic polymer. Wax-like aliphatic compounds are ubiquitous in the ambient atmosphere due to the emissions from vehicle combustions, vegetation, and industries. This wax-like compounds can deposit onto the Teflon film. Teflon film can be exposed to ambient air by numerous ways (manufacturing sites, transportation from the manufacturing site to laboratories or Teflon film bag manufacturers, and ventilation of the chamber with the ambient air after each chamber experiment). Thus, most Teflon film surfaces are contaminated by wax-like materials in the ambient air. Prior to each experiment, the outdoor chamber air is cleaned with a clean air generator under the ambient sunlight for 2-3 days. However, the wax-like matter is not completely removed.

    b. *Extraction method of OM:* It is a good point. We agree with the review's concern about the extraction method. We tested with several solvent (methylene chloride, acetonitrile, and acetone). The resulting spectra are similar even with different solvent. A small quantity of inorganic salts can be appeared with polar solvent, but the main composition is wax-like matter. The evaporation procedure may impact compositions of $OM_{wall}$ but it would be small because the chamber is vented with the clean air and the volatile compounds are evaporated. The large uncertainty in the characterization of $OM_{wall}$ would be the OM extraction efficiency associated with a wiping method. This wax-like material may penetrate into the certain layers of Teflon film near the surface and influence the property of Teflon film. As we mention in Section S2 (Mass concentration of $M_{wall-OM}$ and its molecular weight), the mass of $OM_{wall}$ measured using the wiping method was lower (30%-50%) than that with the solvent extraction by socking into a large amount of several solvents (methylene chloride, acetonitrile, and acetone). In this study, the mass of $OM_{wall}$ was determined with the solvent extraction method and the $MW_{OM}$ of $OM_{wall}$ was determined by using FTIR data associated with the wiping method.

***Comment 2)*** There is increasing evidence that secondary organic aerosols from oxidation of VOCs such as alpha-pinene consist of LVOCs and ELVOCs that contain -OOH functional groups (Bianchi et al., 2019). The authors need to broaden the discussions on the implications/limitations of using the descriptor to estimate gas-wall process and its effects on SOA mass predictions regarding -OOH and (E)LVOCs.

***Response:***

In order to respond to the reviewer, the feasibility of QSAR on the prediction of GWP of the compound containing -OOH was discussed in the revised manuscript.

"There is increasing evidence that SOA from oxidation of the reactive hydrocarbon such as α-pinene contains both low volatility organic products and extremely low volatile organic products with -OOH functional groups (Bianchi et al., 2019). The QSAR-base model of this study is also capable of supporting the prediction of the GWP of the products with –OOH functionality."

*Comment 3)* As a figure in the main text, Figure 3 deserves more description and discussions. The observed time sequences of 1-heptanoic acid and 2,5-dimethylphenol do not show a downward trend as the predicted time sequences at 40% RH. The authors need to provide more explanation.

*Response:*

The observed gas-phase concentration of 1-heptanoic acid and 2,5-dimethylphenol do not show a decreasing trend as the predicted values because they can rapidly reach to equilibrium at 0.4 RH. As discussed in the section 4.3, the longer $\tau_{GWP,i}$ was found for SVOCs with the higher $K_{w,i}$. By reducing volatility, the SVOC has a greater $K_{w,i}$ with a large carbon number or strong hydrogen bonding. Both 1-heptanoic acid and 2,5-dimethylphenol are the compounds with small carbon numbers and strong hydrogen bonding. However, strong hydrogen bonding cannot increase $K_{w,i}$ a lot under the lower RH than 0.5. As seen in Fig. S4, the surface of Teflon film ($OM_{wall}$) is relatively hydrophobic and the water contend in $OM_{wall}$ (or surface Teflon film) is very little before 0.5 RH (Fig. S5). It suggesting that $K_{w,i}$ of the SVOCs with polar functional group is similarly small under lower RH than 0.5. In addition, a longer $\tau_{GWP,i}$ was found for SVOCs with large $\alpha_{w,i}$, which is smaller with increasing molecular size.

*Comment 4)* Line 104: It is confusing here. Are there particles in these experiments or not? Figure 2 indicates there are no particles but line 104 indicates there are particles.

*Response:*

The sentient has been revised and reads now,

"No particle appeared after vaporizing organic chemicals into the chamber air based on particle data, which was monitored using a scanning mobility particle sizer (SMPS, TSI 3080, Shoreview, MN, USA) and a condensation particle counter (CPC, TSI 3022A, Shoreview, MN, USA)."

*Minor comments:*

*Comment 5)* Line 8, PaDEL-Descriptor, a software that calculates. . .

*Response:*

The sentence pointed by the reviewer has been corrected.

*Comment 6)* Table S2 and S3: There are several coefficients that have a p-value greater than

0.05, are those all included as descriptors?

***Response:***

Several descriptors show larger p-value than 0.05. In particular, the high p-value of $E_i$ and $H_{a,i}$ indicates that those are insignificant (p-value $> 0.05$) to the $\ln(\gamma_{w,i})$. The definition of insignificant parameters was considered to determine the reasonable GWP predictive polynomial equation. $E_i$ can partially involve by other terms of the $K_{w,i}$ (Eq. 11) because $E_i$ is calculated based on the molecular weight and density. Thus, $E_i$ was eliminated from the polynomial equation. $H_{a,i}$ was included as a descriptor of the polynomial equation to consider energetic contribution of hydrogen bonding interactions between hydrogen accepting SVOCs and hydrogen donating wall composition. Then, based on the adjusted $R^2$ values in linear regressions, parameters $H_{d,i}$, $H_{a,i}$, $\alpha_i$, and $S_i$ were applied to the derivation of the QSAR-based polynomial equation.

Reference:

Bianchi, Federico, et al. "Highly oxygenated organic molecules (HOM) from gas-phase autoxidation involving peroxy radicals: A key contributor to atmospheric aerosol." Chemical reviews119.6 (2019): 3472-350